# LLaVA-OneVision: Easy Visual Task Transfer

**Bo Li**                                                            *libo0013@e.ntu.edu.sg*
*S-Lab, Nanyang Technological University*

**Yuanhan Zhang**                                          *yuanhan002@e.ntu.edu.sg*
*S-Lab, Nanyang Technological University*

**Dong Guo**                                                  *dongguo.dguo@bytedance.com*
*ByteDance*

**Renrui Zhang**                                              *renruizhang@link.cuhk.edu.hk*
*Chinese University of Hong Kong*

**Feng Li**                                                            *fliay@connect.ust.hk*
*Hong Kong University of Science and Technology*

**Hao Zhang**                                                    *hzhangcx@connect.ust.hk*
*Hong Kong University of Science and Technology*

**Kaichen Zhang**                                              *zhan0564@e.ntu.edu.sg*
*S-Lab, Nanyang Technological University*

**Peiyuan Zhang**                                            *peiyuan.zhang@ntu.edu.sg*
*S-Lab, Nanyang Technological University*

**Yanwei Li**                                                      *liyanwei@link.cuhk.edu.hk*
*ByteDance*

**Ziwei Liu**                                                        *ziwei.liu@ntu.edu.sg*
*S-Lab, Nanyang Technological University*

**Chunyuan Li**                                                  *lichunyuan24@gmail.com*
*ByteDance*

**Reviewed on OpenReview:** *https://openreview.net/forum?id=zKv8qULV6n*

## Abstract

We present LLaVA-OneVision, a family of open large multimodal models (LMMs) developed by consolidating our insights into data, models, and visual representations in the LLaVA-NeXT blog series. Our experimental results demonstrate that LLaVA-OneVision is the first single model that can simultaneously push the performance boundaries of open LMMs in three important computer vision scenarios: single-image, multi-image, and video scenarios. Importantly, the design of LLaVA-OneVision allows strong transfer learning across different modalities/scenarios, yielding new emerging capabilities. In particular, strong video understanding and cross-scenario capabilities are demonstrated through task transfer from images to videos.

# 1 Introduction

It is a core aspiration in AI to build general-purpose assistants with Large Multimodal Models (LMM) (Li et al., 2024c). LLaVA-OneVision is an open model, continuing to advance the line of research in building large vision-and-language assistant (LLaVA) (Liu et al., 2023c) that can follow diverse instructions to complete a variety of computer vision tasks in the wild. As a cost-efficient recipe, it is typically developed by connecting vision encoders with large language models (LLM) using a simple connection module.

The first LLaVA model (Liu et al., 2023c) demonstrates impressive multimodal chat abilities, sometimes exhibiting the behaviors similar to GPT-4V on previously unseen images and instructions for the first time. LLaVA-1.5 (Liu et al., 2024a) significantly expands and improves the capabilities by incorporating more academic-related instruction data, achieving SoTA performance on a dozens of benchmarks with a data-efficient recipe. LLaVA-NeXT (Liu et al., 2024b) inherits this property, further pushing performance boundaries through three key techniques: AnyRes for handling high-resolution images, expanding high-quality instruction data, and utilizing the best open LLM available at the time.

LLaVA-NeXT provides an extendable and scalable prototype, which facilitates several parallel explorations, reported in the LLaVA-NeXT blog series (Liu et al., 2024b; Zhang et al., 2024h; Li et al., 2024b;a;d):

- The *Video* blog (Zhang et al., 2024h) shows that the image-only-trained LLaVA-NeXT model is surprisingly strong on video tasks with zero-shot modality transfer, due to the design of AnyRes to digest any vision signals as a sequence of images.
- The *Stronger* blog (Li et al., 2024b) demonstrates the LLM model scaling succuss of this cost-efficient strategy. By simply scaling up the LLM, it achieves performance comparable to GPT-4V on selected benchmarks.
- The *Ablation* blog (Li et al., 2024a) summarizes our empirical exploration except the visual instruction data itself, including the choice of architectures (scaling of LLM & vision encoder), visual representations (resolution & #tokens), as well as training strategies (trainable modules & high-quality data) in the pursuit of data scaling success.
- The *Interleave* blog (Li et al., 2024d) describes the strategies to extend and improve the capability in new scenarios including multi-image, multi-frame (video) and multi-view (3D), while maintaining the single-image performance.

Motivated by these explorations, we aim to offer useful insights and develop the next state-of-the-art level model to handle versatile visual inputs. During the process, we have also been accumulating and curating a large collection of the high-quality datasets. By consolidating these insights, we introduce LLaVA-OneVision. We implement the new model with the available compute, without extensively de-risking individual components. This leaves room for further improvements in capabilities through additional data and model scaling following our recipe. In particular, our paper makes the following contributions:

- *Large multimodal models.* We develop LLaVA-OneVision, a family of open large multimodal models (LMMs) that improves the performance boundaries of open LMMs in three important vision settings, including single-image, multi-image, and video scenarios.
- *Emerging Capabilities with Task Transfer.* Our design in modeling and data representations allow task transfer across different scenarios, suggesting a simple approach to yield new emgerging capabilities. In particular, LLaVA-OneVision demonstrate strong video understanding through task transfer from images.
- *Open-source.* To pave the way towards building a general-purpose visual assistant, we release the following assets to the public: the generated multimodal instruction data, the codebase, the model checkpoints, and a visual chat demo.

# 2 Related Work

The SoTA proprietary LMMs, such as GPT-4V (OpenAI, 2023), GPT-4o (OpenAI, 2024), Gemini (Team et al., 2023) and Claude-3.5 (Anthropic, 2024), exhibit excellent performance in versertile vision scenarios, including single-image, multi-image and video settings. In the open research community, existing works

typically develop models tailored to each individual scenario separately. Specifically, most focus on pushing the performance limits in single-image scenarios (Dai et al., 2024; Liu et al., 2023c; Zhu et al., 2023; Li et al., 2024g; Zhang et al., 2023a; Guo et al., 2023), only a few recent papers have begun to explore multi-image scenarios (Li et al., 2023a; Jiang et al., 2024). While video LMMs excel in video understanding, they often do so at the expense of image performance (Li et al., 2024f; Lin et al., 2023). It is rare to have a single open model that reports excellent performance in all three scenarios. LLaVA-OneVision aims to fill this gap by demonstrating state-of-the-art performance across a broad range of tasks, and showcasing interesting emerging capabilities through cross-scenario task transfer and composition.

To the best of our knowledge, LLaVA-NeXT-Interleave (Li et al., 2024d) is the first attempt to report good performance in all three scenarios, LLaVA-OneVision inherits its training recipe and data for improved performance. Other versatial open LMMs with potentials to excel include VILA (Lin et al., 2024), InternLM-XComposer-2.5 (Zhang et al., 2024c). Unfortunately, their results are not fully evaluated and reported; we compare with them in the experiments. In addition to building systems with versatial capabilities, LLaVA-OneVision is benefited from large-scale high-quality data training, including model-synthesized knowledge and the new collection of diverse instruction tuning data. For the former, we inherit all the knowledge learning data in (Li et al., 2024a). For the latter, our are motivated by FLAN (Wei et al., 2021; Longpre et al., 2023; Xu et al., 2024c). The data collection process is con-current with Idefics2 (Laurençon et al., 2024) and Cambrian-1 (Tong et al., 2024), but we focus on a smaller but more carefully curated collection of datasets. A similar conclusion is observed: a large amount of visual instruction tuning data can significantly improve performance. For comprehensive investigations on design choices of LMMs, we refer to several recent studies (Karamcheti et al., 2024; Laurençon et al., 2024; Li et al., 2024a; McKinzie et al., 2024; Tong et al., 2024; Beyer et al., 2024).

## 3 Modeling

### 3.1 Network Architecture

The model architecture inherits the minimalism design of LLaVA series, whose primary goals are (*i*) effectively leverage the pre-trained capabilities of both the LLM and visual model, as well as (*ii*) facilitate strong scaling behavior in terms of both data and model. The network archtecture is illustrated in Figure 1.

- *LLM.* We choose Qwen-2 (Yang et al., 2024) as our LLM $f_\phi(\cdot)$ parameterized by $\phi$, as it offers various model size and exhibits strong language capabilities to date among publicly available checkpoints.
- *Vision Encoder.* We consider the SigLIP (Zhai et al., 2023) as the visual encoder $g_\psi(\cdot)$ parameterized by $\psi$, encoding an input image $\mathbf{X}_\mathrm{v}$ into its visual feature $\mathbf{Z}_\mathrm{v} = g(\mathbf{X}_\mathrm{v})$. The grid features before and after the last Transformer layer are considered in our experiments.
- *Projector.* We consider a 2-layer MLP (Liu et al., 2024a) $p_\theta(\cdot)$ parameterized by $\theta$, to project image features into the word embedding space, yielding a sequence of visual tokens $\mathbf{H}_\mathrm{v} = p(\mathbf{Z}_\mathrm{v})$.

The model choice is based on our empirical insights in (Li et al., 2024b;a) that stronger LLM typically supercharge stronger multimodal capabilities in the wild, while SigLIP yields higher LMM performance among open vision encoders.

For a sequence of length $L$, we compute the probability of the target answers $\mathbf{X}_\mathrm{a}$ by:

$$p(\mathbf{X}_\mathrm{a}|\mathbf{X}_\mathrm{v}, \mathbf{X}_\mathrm{q}) = \prod_{i=1}^{L} p(\boldsymbol{x}_i|\mathbf{X}_\mathrm{v}, \mathbf{X}_{\mathrm{q},<i}, \mathbf{X}_{\mathrm{a},<i}), \tag{1}$$

where $\mathbf{X}_{\mathrm{q},<i}$ and $\mathbf{X}_{\mathrm{a},<i}$ are the instruction and answer tokens in all turns before the current prediction token $\boldsymbol{x}_i$, respectively. For the conditionals in equation 1, we explicitly add $\mathbf{X}_\mathrm{v}$ to emphasize the fact that the visual signal is grounded for all answers. As explained in Section 3.2, the form of visual signal $\mathbf{X}_\mathrm{v}$ is general. The visual input fed into the vision encoder depends on the corresponding scenarios: the invidiual image crop in the single-image sequence, the invidiual image in a multi-image sequence and the invidiual frame in the video sequence, respectively.

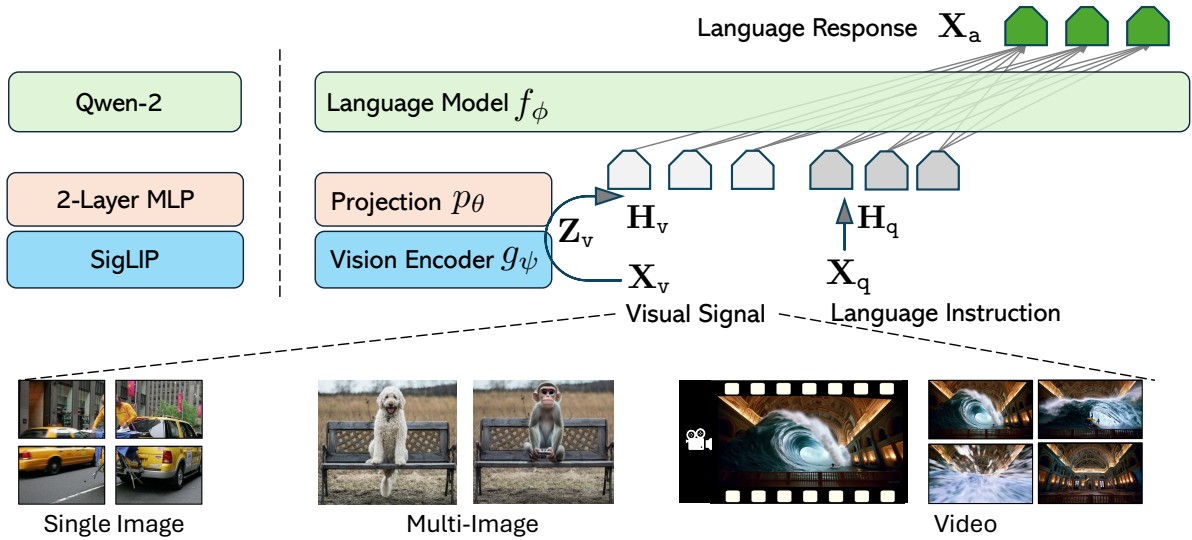

Figure 1: LLaVA-OneVision network architecture. Left: The current model instantiation; Right: the general form of LLaVA architecture in (Liu et al., 2023c), but is extended to support more visual signals.

## 3.2 Visual Representations

The representation of visual signals is key to the success of the visual encoding. It relates to two factors, *the resolution in the raw pixel space* and *the number of tokens in the feature space*, leading to the visual input representation configuration (resolution, #token). The scaling of both factors leads to improved performance, especially on tasks that require visual details. To strike a balance of performance and cost, we observe that the scaling of resolution is more effective than that of token numbers, and recommend an AnyRes strategy with pooling. The comparison is illustrated in Figure 2.

For AnyRes with a configuration of width $a$, height $b$, it divides the image into $a \times b$ crops, each with the shape $(a, b)$. Each crop has the same resolution suitable for the vision encoder. Assuming there are $T$ tokens per crop, the total number of visual tokens is $L = (a \times b + 1) \times T$, where the base image is resized before being fed into the vision encoder. We consider a threshold $\tau$, and reduce the #token per crop, using bilinear interpolation if needed:

$$T_{\text{new}} = \begin{cases} \frac{\tau}{(a \times b + 1)} & \text{if } L > \tau \\ T & \text{if } L \leq \tau \end{cases} \qquad (2)$$

A set of spatial configurations $(a, b)$ is defined to specify various methods for cropping images, thereby accommodating images of different resolutions and aspect ratios. Among them, the configuration that requires a minimum number of crops is selected. Please see our detailed ablations of visual representation in (Li et al., 2024a).

The proposed Higher AnyRes strategy can serve as a flexible visual representation framework, adaptable for multi-image and video representation. The optimal configuration for performance and cost can be adjusted accordingly. We illustratie the configuration in Figure 3, describe the detailed in Section A.1 and provide high-level encoding strategies as below:

- *Single-image.* We consider a large maximum spatial configuration $(a, b)$ for single-image representation to maintain the original image resolution without resizing. Additionally, we purposefully allocate a large number of visual tokens per image, resulting in a long sequence to effectively represent the visual signal. This is based on the observation that there is a larger number of high-quality training samples with diverse instructions for images compared to videos. By representing an image with a long sequence that mimics video representation, we facilitate a smoother capability transfer from image to video understanding (Zhang et al., 2024h; Li et al., 2024a).

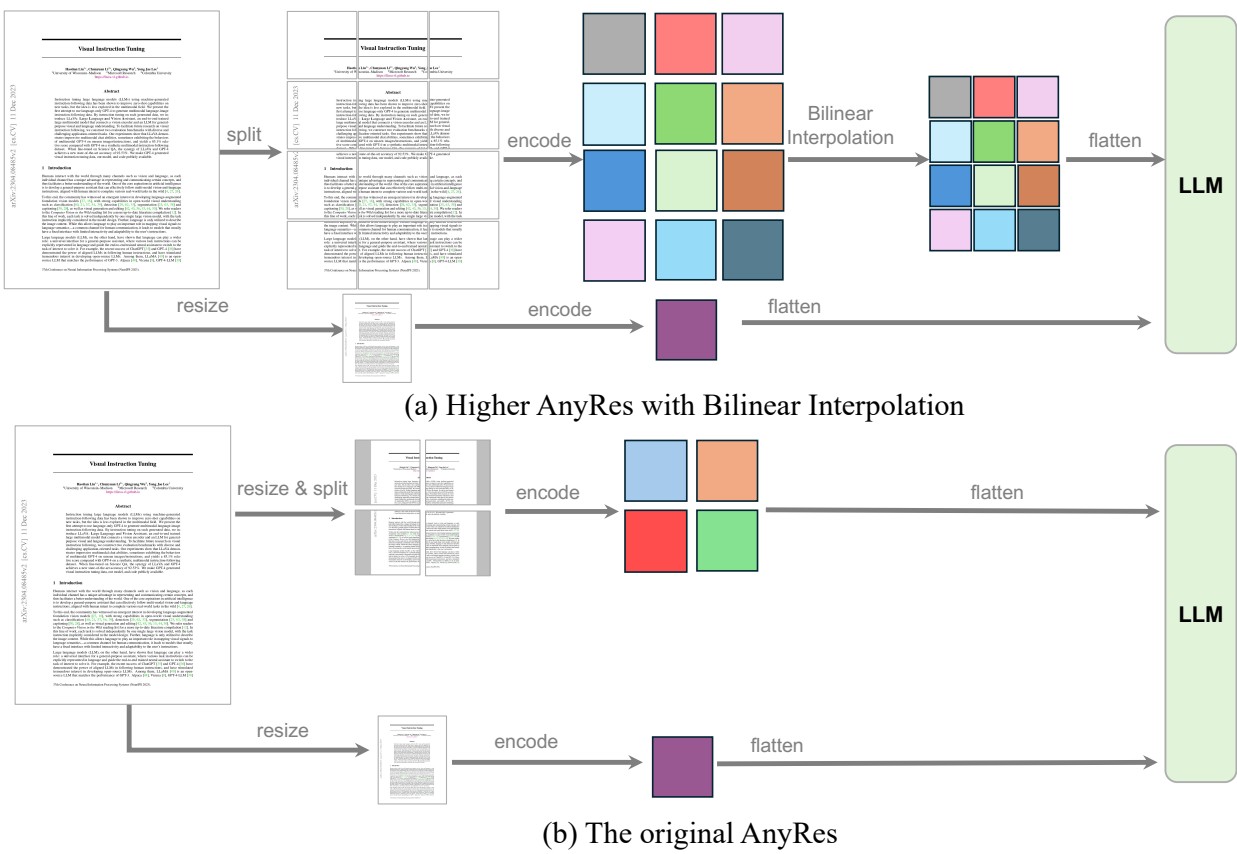

Figure 2: The visual representations. Top: The new Higher AnyRes scheme with Bilinear Interpolation to deal with images of higher resolution; Bottom: the original AnyRes in (Liu et al., 2024b).

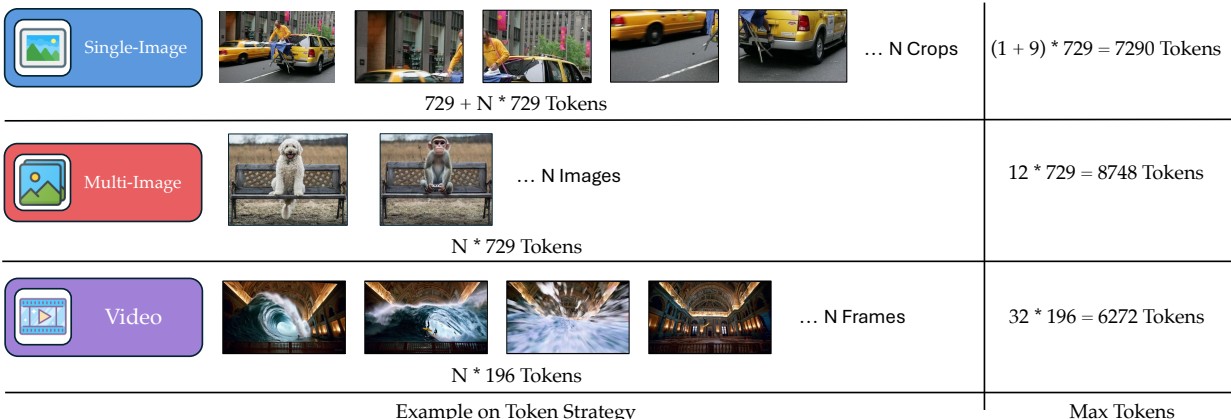

Figure 3: The visual representation strategy to allocate tokens for each scenario in LLaVA-OneVision. The maximum number of visual tokens across different scenarios is designed to be similar, ensuring balanced visual representations to accommodate cross-scenario capability transfer. Note that 729 is the #tokens for SigLIP to encode a visual input of resolustion 384×384.

- *Multi-image.* Only the base image resolution is considered and fed into the vision encoder to obtain feature maps, eliminating the need for multi-crop of high resolution image and thus saving computational resources (Li et al., 2024d).

- *Video.* Each frame of the video is resized to the base image resolution and processed by the vision encoder to generate feature maps. Bilinear interpolation is employed to reduce the number of tokens, allowing the consideration of a larger number of frames by reducing tokens per frame. Empirical evidence suggests this provides a better trade-off between performance and computational cost (Zhang et al., 2024h).

These representation configurations are designed for capability transfer with a fixed compute budget in our experiments. With increased computational resources, the number of tokens per image or frame can be increased during both training and inference stages to boost performance.

# 4 Data

In the realm of multimodal training from LLM, the axiom "quality over quantity" is especially true. This principle is paramount due to the extensive knowledge stored within pre-trained LLMs and Vision Transformers (ViTs). While it is essential to accumulate balanced, diverse, and high-quality instruction data by the end of the LMM's training lifecycle, an often-overlooked aspect is the continuous exposure of the model to new, high-quality data for further knowledge acquisition whenever it is available. In this section, we discuss the data sources and strategies for high-quality knowledge learning and visual instruction tuning.

## 4.1 High-Quality Knowledge

The web-scale public image-text data is often of low-quality, rendering the data scaling of multimodal pre-training less efficient. Instead, we recommend to focus on high-quality knowledge learning, given a limited compute budget. This approach acknowledges that the pre-trained LLMs and ViTs already possess a substantial knowledge base, and the goal is to refine and enhance this knowledge with carefully curated data. By prioritizing the quality of data, we can maximize compute efficiency.

We consider data from three major categories for high-quality knowledge learning:

- *Re-Captioned Detailed Description Data.* LLaVA-NeXT-34B (Liu et al., 2024b) is known for its strong detailed caption ability among open-source LMMs. We used the model to generate new captions for the images from the following datasets: COCO118K, BLIP558K, and CC3M. We combined them to form the Re-Captioned Detailed Description Data, totaling 3.5M samples. This can be viewed as an simple attempt of self-improvement AI, where the training data is generated by an early version of the model itself.
- *Document / OCR Data.* We utilized the Text Reading subset from the UReader dataset, totaling 100K, which is easily accessible through PDF rendering. We used this text reading data along with the SynDOG EN/CN, to form the Document / OCR Data, totaling 1.1M samples.
- *Chinese and Language Data.* We used the original ShareGPT4V (Chen et al., 2023a) images and utilized GPT-4V provided by the Azure API to generate 92K detailed Chinese caption data, aiming to improve the model's capability in Chinese. Since we used a large portion of detailed caption data, we also aim to balance the model's language understanding ability. We collected 143K samples from the Evo-Instruct dataset (Chen et al., 2024a).

It is interesting to note that almost all (accounting for 99.8%) of the high-quality knowledge data is synthetic. This is due to the high cost and copyright constraints associated with collecting large-scale, high-quality data in the wild. In contrast, synthetic data can be easily scaled. We believe that learning from large-scale synthetic data is becoming a trend as AI models continue to grow more powerful.

## 4.2 Visual Instruction Tuning Data

Visual instruction tuning (Liu et al., 2023c) refers to the capability of an LMM to understand and act upon visual instructions. These instructions can be in the form of language, combined with visual media such as images and videos, which the LMM processes and follows to perform a task or provide a response. This involves integrating visual understanding with natural language processing to interpret the instructions and execute the required responses.

**Data Collection and Curation.** As demosntrated in previous works (Liu et al., 2024a; Tong et al., 2024; Laurençon et al., 2024), visual instruction tuning data is crutial for LMM capaiblity. Therefore, maintaining a high-quality dataset collection is crucial and beneficial to the community. We started to collect a large pool of instruction tuning datasets from various original sources, with an unbalanced data ratio among categories. Additionally, we utilize a few new subsets from the Cauldron (Laurençon et al., 2024) and Cambrian (Tong et al., 2024) dataset collections.

We categorize the data based on a three-level hierachy: vision, instruction, and response.

- *Vision Input.* Three vision scenarios are considered, depding which visual input is considered in the multimodal sequence, including single-image, multi-image, video.
- *Language Instruction.* The instructions, which often appears as questions, define the tasks to perform to deal with the visual input. We classify the data into five major categories: *General QA*, *General OCR*, *Doc/Chart/Screen*, *Math Reasoning*, and *Language*. These instructions define the skill sets that a trained LMM could cover. We use task categorization to help maintain and balance the skill distribution.
- *Language Response.* The answer not only responds the user request, but also specifies the model behavior. It can be broadly categorized into free-form and fixed-form.

Free-form data is typically annotated by advanced models like GPT-4V/o and Gemini, while fixed-form data is derived from academic datasets, e.g. VQAv2, GQA, Visual Genome. For free-form data, we keep the original answers. However, for fixed-form data, we manually review the content and make necessary corrections to the question and answer formats. We adhere to the LLaVA-1.5 prompting strategy for multiple-choice data, short answer data, and specific task data (e.g., OCR). This step is crucial for guiding the model's behavior to correctly balance QA performance, conversational ability, and reasoning skills in more complicated tasks, as well as preventing potential conflicts from different data sources. We list the full details about each dataset in our collection, and their categorization and formatting prompt in Appendix C.3.

We divide the instruction data into two separate groups: one for single-image scenario and the other for all vision scenarios. This division is based on insights from our earlier studies (Li et al., 2024d; Zhang et al., 2024h), which highlight the relationship between image and video models: a stronger image model can better transfer to multi-image and video tasks. Additionally, the quantity and quality of training datasets available for single images are significantly higher than those for videos and multi-image tasks.

**Single-Image Data.** Since single-image data is crucial for multimodal capabilities, we explicitly compile a large single-image data collection for model learning. We select from collected data sources to form a balanced collection, resulting in a total of 3.2 million samples. The overall distribution of single-image data is shown in Figure 4, with detailed information and the roadmap of data collection presented in Appendix C.1.

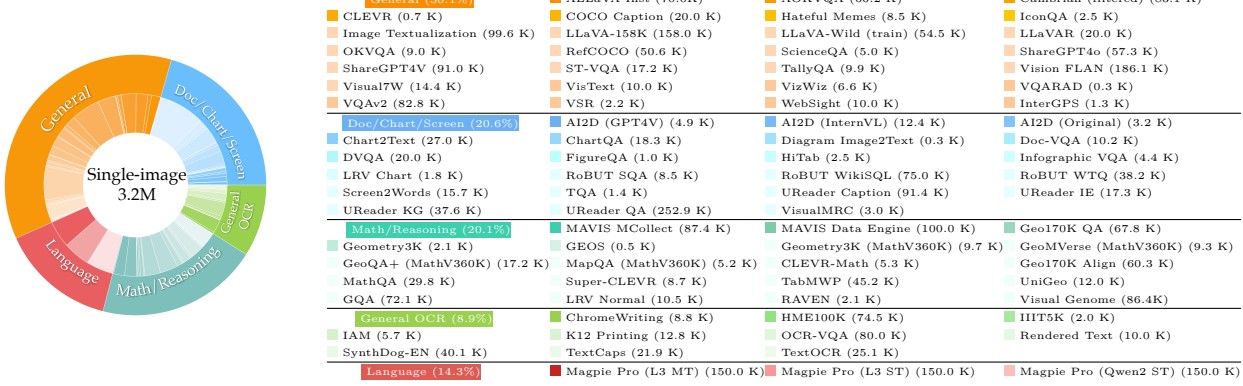

Figure 4: **Single-Image 3.2M.** A High-Quality Single-Image Dataset Collection. Left: Data Distribution within Each Category. The outer circle shows the distribution of all data categories and the inner circle shows the distribution of data subsets. Right: The detailed quantities of datasets.

**OneVision Data.** In addition to the single-image stage training, we further fine-tune the model using a mixture of video, image, and multi-image data. We introduce a total of 1.6 million mixed data samples,

comprising 560K multi-image data from Li et al. (2024d), 350K videos collected in this project, and 800K single-image samples. Notably, in this stage, we do not introduce new single-image data but instead sample high-quality and balanced portions from the previous single-image data, as described in Li et al. (2024d). The data distribution and details are presented in Figure 5, with additional information available in Appendix C.2.

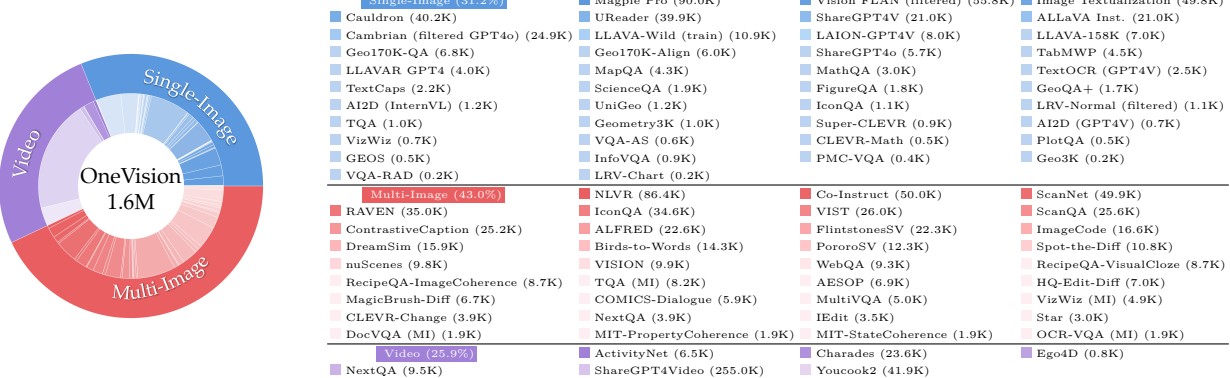

Figure 5: **OneVision 1.6M.** A high-quality single-image, multi-image and video dataset collection. Left: Data Distribution within each category. The outer circle shows the distribution of all data categories and the inner circle shows the distribution of data subsets. Right: The detailed quantities of datasets. "MI" means it is the multi-image version dataset proposed by DEMON (Li et al., 2024e).

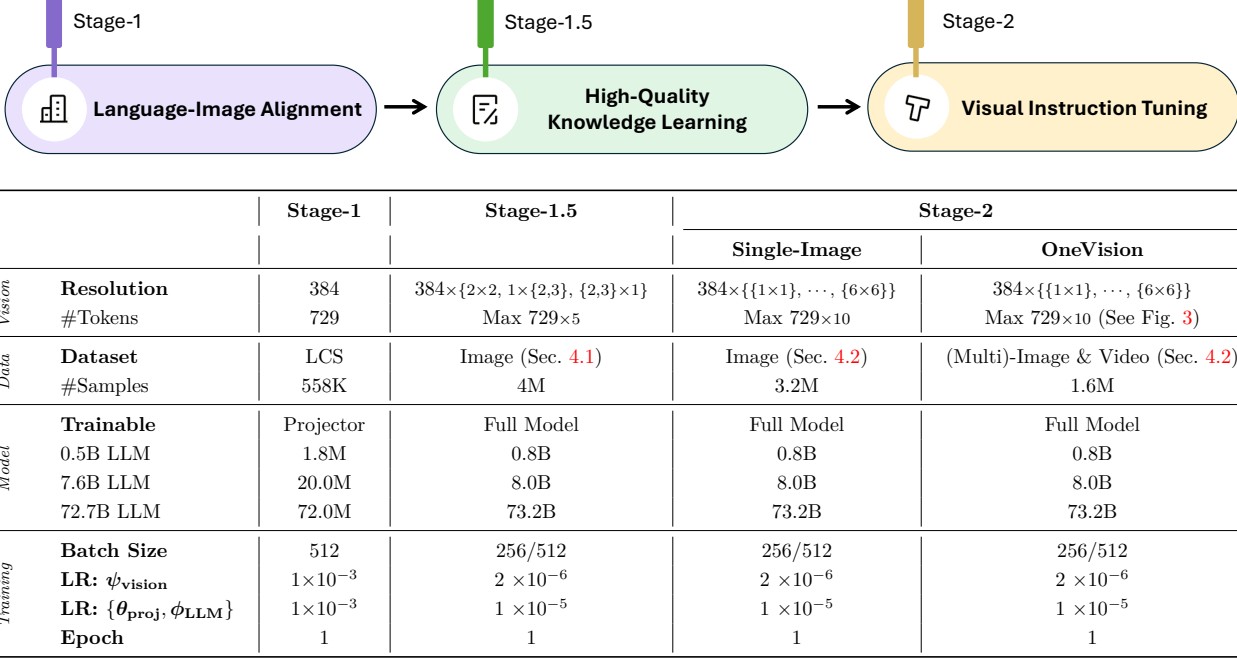

| | | Stage-1 | Stage-1.5 | Stage-2 | |
| | | | | **Single-Image** | **OneVision** |
|---|---|---|---|---|---|
| *Vision* | **Resolution** | 384 | 384×{2×2, 1×{2,3}, {2,3}×1} | 384×{{1×1}, ⋯, {6×6}} | 384×{{1×1}, ⋯, {6×6}} |
| | #Tokens | 729 | Max 729×5 | Max 729×10 | Max 729×10 (See Fig. 3) |
| *Data* | **Dataset** | LCS | Image (Sec. 4.1) | Image (Sec. 4.2) | (Multi)-Image & Video (Sec. 4.2) |
| | #Samples | 558K | 4M | 3.2M | 1.6M |
| *Model* | **Trainable** | Projector | Full Model | Full Model | Full Model |
| | 0.5B LLM | 1.8M | 0.8B | 0.8B | 0.8B |
| | 7.6B LLM | 20.0M | 8.0B | 8.0B | 8.0B |
| | 72.7B LLM | 72.0M | 73.2B | 73.2B | 73.2B |
| *Training* | **Batch Size** | 512 | 256/512 | 256/512 | 256/512 |
| | **LR:** $\psi_{\mathbf{vision}}$ | $1\times10^{-3}$ | $2\times10^{-6}$ | $2\times10^{-6}$ | $2\times10^{-6}$ |
| | **LR:** $\{\theta_{\mathbf{proj}}, \phi_{\mathbf{LLM}}\}$ | $1\times10^{-3}$ | $1\times10^{-5}$ | $1\times10^{-5}$ | $1\times10^{-5}$ |
| | **Epoch** | 1 | 1 | 1 | 1 |

Table 1: Detailed configuration for each training stage of the LLaVA-OneVision model. The table outlines the progression of vision parameters, dataset characteristics, model specifications, and training hyperparameters across different stages of the curriculum learning process. We use a global batch size of 512 for the 0.5B model, and 256 for the 7B and 72B models.

## 5 Training Strategies

To enable LLM for multimodal capabilities, we identify three critical functionalities, and systematically divide them into three distinct learning stages for the purpose of ablation studies. As with most existing

research, prior LLaVA models mainly explore the single-image instruction tuning. However, other parts are less frequently investigated and therefore constitute the primary focus of this section.

We train the model via a curriculum learning principle, where training objectives and examples of increasing difficulty are observed in a stage-wise manner. With a fixed compute budget, this strategy helps decompose the training process and produces immediate checkpoints that can be re-used in more experiment trails.

- *Stage-1: Language-Image Alignment.* The goal is to well align the visual features into the word embedding space of LLMs.
- *Stage-1.5: High-Quality Knowledge Learning.* To strike a balance between compute-efficiency and injecting new knowledge into LMMs, we recommend to consider the high-quality knowledge for LMM learning. The training configuration mirrors the settings used in Stage-2, ensuring consistency and allowing the model to integrate new information seamlessly.
- *Stage-2: Visual Instruction Tuning.* To teach LMM to solve a diverse set of visual task with preferred responces, we organize the instruction data into different groups, described in Section 4.2. The model is scheduled to train on these groups in order.

Specifically, the visual instruction tuning process consists of two phases: (*i*) *Single-Image Training*: The model is first trained on 3.2 million single-image instructions, resulting in a model with strong performance in following a diverse set of instructions to complete visual tasks using a single image. (*ii*) *OneVision Training*: The model is then trained on a mixture of video, single-image, and multi-image data. In this phase, the model expands its capabilities from single-image scenarios to diverse scenarios. It learns to follow instructions to complete tasks in each new scenario and transfer the learned knowledge across different scenarios, resulting in new emergent capabilities. Note that the proposed OneVision training in the post-training stage is probably the simplest and most cost-efficient way to empower the LMMs with the multi-image and video understanding capabilities.

The training strategy is summarized in Table 1. We progressively train the model to deal with long sequence training. The maximum image resolution and the number of visual tokens gradually increase as training progresses. In Stage-1, the base image representation is considered with 729 tokens. In Stages 1.5 and 2, AnyRes is considered with up to 5 times and 10 times more visual tokens, respectively. Regarding trainable modules, Stage-1 updates only the projector, while the subsequent stages update the full model. It is also noted that the learning rate for the vision encoder is 5 times smaller than that for the LLM.

# 6 Experimental Results

We conduct standardized and reproducible evaluations for LLaVA-OneVision models on all benchmarks using LMMs-Eval (Zhang et al., 2024b). For fair comparison with other leading LMMs, we primarily report results from original papers. When results are unavailable, we onboard the models in LMMs-Eval and evaluate them using consistent settings. All our results are reported with greedy decoding and 0-shot settings unless otherwise specified.

To reveal the generality and effectiveness of the designed paradigm, we comprehensively evaluate our LLaVA-OneVision models across different modalities in Table 2, including single-image, multi-image, and video benchmarks. Detailed results for each modality are presented in Table 3, Table 4, and Table 5, respectively. We denote the the model checkpoint trained after the single-image stage and one-vision stage as *LLaVA-OV (SI)* or *LLaVA-OV*, respectively

Three model sizes are provided (0.5B, 7B and 72B), to accomodate applications with different performance-throughput trade-off, ranging from edge device to cloud serving. The GPT-4V and GPT-4o results are presented as references. Our largest model LLaVA-OneVision-72B yields superior performance between GPT-4V and GPT-4o on most benchmarks. It suggests that the proposed recipe is effecitve, revealing a promising path for further scaling. However, a relatively larger gap remains in complex tasks such as visual chat scenarios, we leave it as future research in stronger LLMs, larger training data and better preference learning.

## 6.1 Single-Image Benchmarks

To validate the performance for single-image tasks in real-world scenories, we consider a comprehensive set of image benchmarks in Table 3. It can be categorized into three classes:

*(1) Chart, Diagram, and Document Understanding.* As the main visual formats for structured OCR data, we evaluate the results on AI2D (Kembhavi et al., 2016a), ChartQA (Masry et al., 2022), DocVQA (Mathew et al., 2021), and InfoVQA (Mathew et al., 2022) benchmarks. Though current open-source models such as InternVL (Chen et al., 2023b) and Cambrian (Tong et al., 2024) achieve performance comparable to commercial models, LLaVA-OneVision goes a step further, surpassing GPT-4V (OpenAI, 2023) and approaching the performance level of GPT-4o (OpenAI, 2024).

*(2) Perception and Multi-discipline Reasoning.* Including visual perception scenarios, we reveal the potentials of our model for more complex and challenging reasoning tasks. Specifically, we adopt the perception benchmarks including MME (Yin et al., 2023), MMBench (Liu et al., 2023e), and MMVet (Yu et al., 2023), and reasoning benchmarks such as MathVerse (Zhang et al., 2024e), MathVista (Lu et al., 2023a), and MMMU (Yue et al., 2024). The results of LLaVA-OneVision significantly outperforms GPT-4V on various benchmarks, and comparable to GPT-4o on MathVista. This further confirms the superiority of our framework in visual perception and reasoning tasks.

*(3) Real-world Understanding and Visual Chat.* We consider the evaluation of LMMs as general-purpose assistant in the wild as the most important metrics, beyond the lab environments. To validate the capabilities in real-world scenarios, we utilize several widely-adopted benchmarks, including RealworldQA (x.ai), Vibe-Eval (Padlewski et al., 2024b), MM-LiveBench (Zhang et al., 2024b), and LLaVA-Bench-Wilder (Li et al., 2024b). While our model still has room for improvement compared to GPT-4V and GPT-4o, it achieves competitive performance with open-source models of similar parameter size. Notably, our model performs well on MM-LiveBench (Zhang et al., 2024b), a benchmark for real-world internet content with constantly updated content, demonstrating the model's broad world knowledge and strong generalization abilities.

## 6.2 Multi-Image Benchmarks

We further evaluate LLaVA-OneVision in multi-image interleaved settings, where users may ask questions between multiples images. In particular, we perform comprehensive assessment on the diverse subtasks of LLaVA-Interleave Bench (Li et al., 2024d), such as Spot the Difference (Jhamtani & Berg-Kirkpatrick, 2018a), Image Edit Instruction (IEI) (Li et al., 2024d), Visual Storytelling (VST) (Huang et al., 2016), Text-rich VQA (TR-VQA) (Liu et al., 2023d), Multi-image VQA (MI-VQA) (Raj et al., 2021), Raven Puzzle (Chia et al., 2024), Q-Bench (QB) (Wu et al., 2023), and NLVR2 (Suhr et al., 2017)). We also utilize several multi-view benchmarks for evaluation, which depict 3D environments with multiple viewpoints, including 3D Dialogue (3D-Chat) and Task Decomposition (3D-TD) from 3D-LLM (Hong et al., 2023), ScanQA (Azuma et al., 2022a), ALFRED (Shridhar et al., 2020), and nuScenes VQA (Bansal et al., 2020). We refer to these datasets as in-domain evaluations, since our training data includes the training split of them.

Moreover, we conduct evaluations on different out-domain tasks, which reveals the generalization capability of our approach. They include the multi-image split of math QA benchmark MathVerse (Zhang et al., 2024e) and science QA benchmark SciVerse (Guo et al., 2024), multi-image perception benchmark BLINK (Fu et al., 2024c), MMMU-(multi-image) (Yue et al., 2024) that contains all multi-image QA in MMMU, and MuirBench (Wang et al., 2024) spanning 12 diverse multi-image tasks.

As shown in Table 4, LLaVA-OneVision (SI) consistently outperforms existing multi-image LMMs in all benchmarks. After additional tuning on multi-image and video data, LLaVA-OneVision shows a marked improvement over GPT-4V in specific areas, with significant margins. This highlights its strong performance in complex tasks such as multi-image reasoning, identifying differences, and understanding 3D environments. In addition, we observe a consistent performance enhancement on after the one-vision training stage, which is more evident on multi-view benchmarks that are absent in single-image data. This demonstrates the significance of our one-vision paradigm for empowering LMMs with comprehensive visual capbalities.

### 6.3 Video Benchmarks

Video is also a common modality to build world model, capturing the dynamic nature of the real world over time. We conduct experiments on several open-ended and multi-choice video benchmarks. These include ActivityNet-QA (Yu et al., 2019) that contains human-annotated action-related QA pairs derived from ActivityNet dataset, EgoSchema (Mangalam et al., 2024) and MLVU (Zhou et al., 2024a) focusing on long video understanding, PerceptionTest (Pătrăucean et al., 2023) designed to evaluate the perception skills, VideoMME (Fu et al., 2024b) and NeXTQA (Xiao et al., 2021) containing diverse video domains and durations (from minutes to hours), VideoDetailCaption (LMMs-Lab, 2024) and Video-ChatGPT (Maaz et al., 2023) for video detailed description and visua chat, respectively.

As shown in Table 5, LLaVA-OneVision achieves comparable or better results than previous open source models with much larger LLMs. The superiority of LLaVA-OneVision is particularly evident in complex benchmarks such as EgoSchema and VideoMME. Even compared to the advanced commercial model GPT-4V, LLaVA-OneVision performs competitively on the ActivityNet-QA, MLVU, and VideoMME benchmarks.

Within the LLaVA-OV split, the smallest performance difference occurs in PerceptionTest, with a minimal improvement of 0.5 points when scaling the LLM from 0.5B to 7B. This contrasts with at least a 5-point improvement in other datasets. The modest gain at PerceptionTest suggests that LLaVA-OV's perception capabilities may mainly depend on its vision module, supporting findings from recent studies such as those by Qiao et al. (Qiao et al., 2024), which separate the roles of the image encoder and the LLM in perception and reasoning tasks. Notably, for datasets like EgoSchema that demand significant reasoning, a larger LLM substantially enhances performance.

Moreover, in comparing LLaVA-OV-7B (SI) with LLaVA-OV-7B, the smallest improvement is seen with ActivityNet-QA. This suggests that LLaVA-OV-7B (SI), which is trained only on images, can already perform well on this dataset. Delving into ActivityNet-QA, it becomes apparent that many questions can be answered by observing just a single frame from the video. For instance, the question "What's the color of the ball?" can be answered throughout the video as the ball is visible from start to finish. This scenario does not require the model to understand the video sequence, allowing LLaVA-OV-7B (SI) to perform well.

## 7 Emerging Capabilities with Task Transfer

In addition to reporting the LLaVA-OneVision's capabilities across various benchmarks, we also observe the emerging behaviors of the proposed model with task transfer and composition, paving a promising way to generalize to tackle real-world computer vision tasks in the wild. We illustrate several emerging capabilities using examples as below.

**S1: Joint understanding of diagram and chart (Transfer from single-image to multi-image)** The capability to understand tables and charts are seperately learned from single image diagram and single-image chart understanding data, and the joint understanding task of table and chart do not appear in multi-image data. As shown in Table 6, LLaVA-OneVision is capable of understanding and reasoning over the joint of diagram and chart.

**S2: GUI for multi-modal agent (Transfer from single-image and multi-image).** Understanding GUIs and applying multimodal models to agentic tasks is of great value. In Table 7, LLaVA-OneVision recognizes the graphical user interface (GUI) screenshots of an iPhone and provides operational instructions to search for and open the TikTok app. This task requires strong OCR capabilities learned from single-image scenarios and relational reasoning skills developed from multi-image scenarios. The example highlights LLaVA-OneVision's proficiency in GUI understanding and task execution.

**S3: Set-of-mark Prompting (Transfer from single-image task composition).** Different from existing open LLMs, LLaVA-OneVision demonstrates excellent set-of-marks (SoM) reasoning (Yang et al., 2023), an emerging capability shown in Table 8. To the best of our knowledge, this is the first time that open LMMs report good emerged SoM ability, as we observe that LLaVA-OneVision is able to produce SoM reasoning for many examples in (Yang et al., 2023). This task is not explicitly included in our training data, it is hypothsized that the ability is composed by visual referring and OCR.

**S4: Image-to-Video Editing Instruction (Transfer from single-image and video).** LLaVA-OneVision could generate detailed video creation prompts based on a static image in Table 9. Given an image and a target video, the model constructs a coherent and vivid narrative for the video, detailing elements such as characters, actions, background settings, and scene specifics. This task leverages both single-image analysis and video comprehension. It is hypothesized that this ability is generalized from the composition of single-image editing instruction task and video detailed description task.

**S5: Video-to-Video Difference (Transfer from multi-image and video).** Understanding differences in images is a common ability in recent large multimodal models (LMMs), but our models extend this capability to videos. Table 10 showcases LLaVA-OneVision's ability to analyze differences between two video sequences with the same beginning frame but different endings. The model provides a detailed comparison, describing characters, actions, and scene changes. In Table 11, LLaVA-OneVision's describe the differences one by one between videos with a similar background but different main object in the foreground. This task leverages spot the difference in the multi-image analysis to generalize to video scenarios.

**S6: Multi-camera Video Understanding in Self-driving (Transfer from single-image and multi-image to video).** Understanding videos in a normal aspect ratio is straightforward, what about the videos with multi-views? In Table 12, we observe that LLaVA-OneVision could analyze and interprets multi-camera video footage from self-driving cars. Given video showing four camera views, the model describes each view in detail and plans the ego car's next move. This task combines multi-panel comprehension, video detailed description, and spatial-temporal reasoning.

**S7: Composed Sub-video Understanding (Transfer from multi-image to video).** Besides multi-view video, we see our model generalize to vertical videos with two sub-scenes. Table 13 demonstrates LLaVA-OneVision's ability to understand and describe the content and layout of a composed sub-video. Given a vertical video with a series of frames featuring a consistent background and a person in the foreground, the model provides a detailed analysis of visual elements, their arrangement, and the narrative context. This task requires single-image analysis, multi-image sequence comprehension, and contextual reasoning.

**S8: Visual prompting in video (Task transfer from single-image to video).** In Table 14, LLaVA-OneVision is able to understand the highlighted area with a semi-transparent circle in the video, and clearly see the number "10" on the back of the player. The capability of understanding visual prompts and OCR is a capablity of single-image LMMs. Our model displays the capablity of understanding visual prompts in videos, without training on video data with visual prompts.

**S9: Visual Referring in Image in Video Understanding.** The ability to refer to image query when answering questions about a video as shown in Table 15. This capbility is not seen in LLaVA-NeXT or LLaVA-Interleave, this is proabably because strong base single-image training is required for such capabilty to appear.

## 8 Conclusions

LLaVA-OneVision is a new, open LMM that shines when transferred to a broad range of tasks in the scenarios of single-image, multi-image and videos. Our design allows new capabilities to emerge, through training multiple scenarios together and task transfer, eg, strong visual understanding ability from image to video. Our results demonstrate that LMMs trained with this open recipe and resources achieve state-of-the-art performance across various benchmarks. We also hope that LLaVA-OneVision serves as a valuable starting point for the community to build specific applications, and develop stronger LMMs for diverse vision scenarios through further scaling.

| Capability | Benchmark | LLaVA OneVision-0.5B | LLaVA OneVision-7B | LLaVA OneVision-72B | GPT-4V (V-Preview) | GPT-4o |
|---|---|---|---|---|---|---|
| Single-Image | †AI2D
Science Diagrams | 57.1% | 81.4% | 85.6% | 78.2% | 94.2% |
| | †ChartQA
Chart Understanding | 61.4% | 80.0% | 83.7% | 78.5% | 85.7% |
| | †DocVQA (test)
Document Understanding | 70.0% | 87.5% | 91.3% | 88.4% | 92.8% |
| | †InfoVQA (test)
Infographic Understanding | 41.8% | 68.8% | 74.9% | - | - |
| | MathVerse (vision-mini)
Professional Math Reasoning | 17.9% | 26.2% | 39.1% | 32.8% | 50.2% |
| | MathVista (testmini)
General Math Understanding | 34.8% | 63.2% | 67.5% | 49.9% | 63.8% |
| | MMBench (en-dev)
Multi-discip | 52.1% | 80.8% | 85.9% | 75.0% | - |
| | MME (cog./perp.)
Multi-discip | 240/1238 | 418/1580 | 579/1682 | 517/1409 | - |
| | MMStar
Multi-discip | 37.5% | 61.7% | 66.1% | 57.1% | - |
| | MMMU (val)
College-level Multi-disp | 31.4% | 48.8% | 56.8% | 56.8% | 69.1% |
| | MMVet Multi-discip | 29.1% | 57.5% | 63.7% | 49.9% | 76.2% |
| | SeedBench (image)
Multi-discip; Large-scale | 65.5% | 75.4% | 78.0% | 49.9% | 76.2% |
| | †ScienceQA
High-school Science | 67.2% | 96.0% | 90.3% | 75.7% | - |
| | RealworldQA
Realworld Scenes | 55.6% | 66.3% | 71.9% | 61.4% | - |
| | Vibe-Eval
Chanllenging Cases | 33.8% | 51.7% | 50.7% | 57.9% | 63.1% |
| | MM-LiveBench (2406)
Internet Content | 49.9% | 77.1% | 81.5% | - | 92.4% |
| | LLaVA-Wilder (small)
Realworld Chat | 55.0% | 67.8% | 72.0% | 81.0% | 85.9% |
| Multi-Image | LLaVA-Interleave
Out-domain | 33.3% | 64.2% | 79.9% | 60.3% | - |
| | MuirBench
Comprehensive Multi-image | 25.5% | 41.8% | 54.8% | 62.3% | - |
| | Mantis
Multi-image in the Wild | 39.6% | 64.2% | 77.6% | 62.7% | - |
| | BLINK
Unusual Visual Scenarios | 52.1% | 48.2% | 55.4% | 51.1% | - |
| | †Text-rich VQA
OCR, Webpage, Ducument | 65.0% | 80.1% | 83.7% | 54.5% | - |
| Video | ActivityNetQA
Spatio-Temporal Reasoning | 50.5% | 56.6% | 62.3% | 57.0% | - |
| | EgoSchema
Egocentric Video | 26.8% | 60.1% | 62.0% | - | - |
| | PerceptionTest
Perception and Reasoning | 49.2% | 57.1% | 66.9% | - | - |
| | SeedBench (video)
Multi-discip; Video | 44.2% | 56.9% | 62.1% | 60.5% | - |
| | LongVideoBench (val)
Long Video | 45.8% | 56.3% | 63.2% | 60.7% | 66.7% |
| | MLVU
Long Video | 50.3% | 64.7% | 68.0% | 49.2% | 64.6% |
| | MVBench
Multi-discip | 45.5% | 56.7% | 59.4% | 43.5% | - |
| | VideoChatGPT
Video Conversation | 3.12 | 3.49 | 3.62 | 4.06 | - |
| | VideoMME
Multi-discip | 44.0% | 58.2% | 66.2% | 59.9% | 71.9% |

Table 2: Performance comparison to state-of-the-art commercial models with our LLaVA-OneVision models (0.5B to 72B parameters) across diverse evaluation benchmarks spanning multiple modalities. † indicates that the training set has been observed in our data mixture.

| Model | AI2D | ChartQA | DocVQA | InfoVQA | MathVerse | MathVista | MMBench | MME | MMMU |
|---|---|---|---|---|---|---|---|---|---|
| | test | test | val/test | val/test | mini-vision | testmini | en-dev | test | val |
| Qwen-VL-Max (Bai et al., 2023b) | 79.3 | 79.8 | -/93.1 | - | 23.0 | 51.0 | 77.6 | 2281 | 51.4 |
| Gemini-1.5-Pro (Team, 2024) | 94.4 | 87.2 | -/93.1 | -/81.0 | - | 63.9 | - | - | 62.2 |
| Claude 3.5 Sonnet (Anthropic, 2024) | 94.7 | 90.8 | -/95.2 | 49.7 | - | 67.7 | - | - | 68.3 |
| GPT-4V (OpenAI, 2023) | 78.2 | 78.5* | -/88.4 | - | 32.8 | 49.9 | 75.0 | 517/1409 | 56.8 |
| GPT-4o (OpenAI, 2024) | 94.2 | 85.7 | -/92.8 | - | 50.2 | 63.8 | - | - | 69.1 |
| Cambrian-34B (Tong et al., 2024) | 79.7 | 73.8 | -/75.5 | - | - | 53.2 | 81.4 | - | 49.7 |
| VILA-34B (Lin et al., 2024) | - | - | - | - | - | - | 82.4 | 1762 | 51.9 |
| IXC-2.5-7B (Zhang et al., 2024c) | 81.5 | 82.2 | -/90.9 | -/70.0 | 20.0 | 59.6 | 82.2 | 2229 | 42.9 |
| InternVL-2-8B (Chen et al., 2023b) | 83.8 | 83.3 | -/91.6 | -/74.8 | 27.5 | 58.3 | 81.7 | 2210 | 49.3 |
| InternVL-2-26B (Chen et al., 2023b) | 84.5 | 84.9 | -/92.9 | -/75.9 | 31.3 | 59.4 | 83.4 | 2260 | 48.3 |
| *LLaVA-OV-0.5B (SI)* | *54.2* | *61.0* | *75.0/71.2* | *44.8/41.3* | *17.3* | *34.6* | *43.8* | *272/1217* | *31.2* |
| LLaVA-OV-0.5B | 57.1 | 61.4 | 73.7/70.0 | 46.3/41.8 | 17.9 | 34.8 | 52.1 | 240/1238 | 31.4 |
| *LLaVA-OV-7B (SI)* | *81.6* | *78.8* | *89.3/86.9* | *69.9/65.3* | *26.9* | *56.1* | *81.7* | *483/1626* | *47.3* |
| LLaVA-OV-7B | 81.4 | 80.0 | 90.2/87.5 | 70.7/68.8 | 26.2 | 63.2 | 80.8 | 418/1580 | 48.8 |
| *LLaVA-OV-72B (SI)* | *85.1* | *84.9* | *93.5/91.8* | *77.7/74.6* | *37.7* | *66.5* | *86.6* | *563/1706* | *57.4* |
| LLaVA-OV-72B | 85.6 | 83.7 | 93.1/91.3 | 79.2/74.9 | 39.1 | 67.5 | 85.9 | 579/1682 | 56.8 |

| Model | MMVet | MMStar | S-Bench | S-QA | ImageDC | MMLBench | RealWorldQA | Vibe-Eval | LLaVA-W | L-Wilder |
|---|---|---|---|---|---|---|---|---|---|---|
| | test | test | image | test | test | 2024-06 | test | test | test | small |
| Qwen-VL-Max (Bai et al., 2023b) | - | - | - | - | - | - | - | - | - | - |
| Gemini-1.5-Pro (Team, 2024) | - | - | - | - | - | 85.9 | 70.4 | 60.4 | - | - |
| Claude 3.5 Sonnet (Anthropic, 2024) | 75.4 | - | - | - | - | 92.3 | 59.9 | 66.2 | 102.9 | 83.1 |
| GPT-4V (OpenAI, 2023) | 49.9 | 57.1 | 49.9 | 75.7 | 91.5 | - | 61.4 | 57.9 | 98.0 | 81.0 |
| GPT-4o (OpenAI, 2024) | 76.2 | - | 76.2 | - | 92.5 | 92.4 | 58.6 | 63.1 | 106.1 | 85.9 |
| Cambrian-34B (Tong et al., 2024) | - | - | - | 85.6 | - | - | 67.8 | - | - | - |
| VILA-34B (Lin et al., 2024) | 53.0 | - | 75.8 | - | - | - | - | 81.3 | - | - |
| IXC-2.5-7B (Zhang et al., 2024c) | 51.7 | 59.9 | 75.4 | - | 87.5 | - | 67.8 | 45.2 | 78.1 | 61.4 |
| InternVL-2-8B (Chen et al., 2023b) | 60.0 | 59.4 | 76.0 | 97.0 | 87.1 | 73.4 | 64.4 | 46.7 | 84.5 | 62.5 |
| InternVL-2-26B (Chen et al., 2023b) | 65.4 | 60.4 | 76.8 | 97.5 | 91.0 | 77.2 | 66.8 | 51.5 | 99.6 | 70.2 |
| *LLaVA-OV-0.5B (SI)* | *26.9* | *36.3* | *63.4* | *67.8* | *83.0* | *43.2* | *53.7* | *34.9* | *71.2* | *51.5* |
| LLaVA-OV-0.5B | 29.1 | 37.5 | 65.5 | 67.2 | 83.3 | 49.9 | 55.6 | 33.8 | 74.2 | 55.0 |
| *LLaVA-OV-7B (SI)* | *58.8* | *60.9* | *74.8* | *96.6* | *85.7* | *75.8* | *65.5* | *47.2* | *86.9* | *69.1* |
| LLaVA-OV-7B | 57.5 | 61.7 | 75.4 | 96.0 | 88.9 | 77.1 | 66.3 | 51.7 | 90.7 | 67.8 |
| *LLaVA-OV-72B (SI)* | *60.0* | *65.2* | *77.6* | *91.3* | *91.5* | *84.4* | *73.8* | *46.7* | *93.7* | *72.9* |
| LLaVA-OV-72B | 63.7 | 66.1 | 78.0 | 90.3 | 91.2 | 81.5 | 71.9 | 50.7 | 93.5 | 72.0 |

Table 3: LLaVA-OneVision performance on single-image benchmarks. *GPT-4V reports 4-shot results on ChartQA. All results are reported as 0-shot accuracy.

| Model | IEI | MI-VQA | NLVR2 | Puzzle | Q-Bench | Spot-Diff | TR-VQA | VST | 3D-Chat | 3D-TD | ScanQA | ALFRED | nuScenes | BLINK | Mantis | MathVerse | MuirBench | SciVerse |
|---|---|---|---|---|---|---|---|---|---|---|---|---|---|---|---|---|---|---|
| | | | in-domain multi-image | | | | | | | in-domain multi-view | | | | | out-domain | | | |
| GPT-4V (OpenAI, 2023) | 11.0 | 52.0 | 88.8 | 17.1 | 76.5 | 12.5 | 54.5 | 10.9 | 31.2 | 35.4 | 32.6 | 10.3 | 63.7 | 51.1 | 62.7 | 60.3 | 62.3 | 66.9 |
| LLaVA-N-Image-7B[†] (Liu et al., 2024b) | 13.2 | 39.4 | 68.0 | 9.0 | 51.0 | 12.9 | 59.6 | 10.1 | - | - | - | - | - | 41.8 | 46.1 | 13.5 | - | 12.2 |
| VPG-C-7B (Li et al., 2023a) | 15.2 | 46.8 | 73.2 | 2.4 | 57.6 | 27.8 | 38.9 | 21.5 | - | - | - | - | - | 43.1 | 52.4 | 24.3 | - | 23.1 |
| Mantis-7B (Jiang et al., 2024) | 11.2 | 52.5 | 87.4 | 25.7 | 69.9 | 17.6 | 45.2 | 12.5 | 2.60 | 14.7 | 16.1 | 14.0 | 46.2 | 46.4 | 59.5 | 27.2 | 36.1 | 29.3 |
| LLaVA-N-Inter-7B (Li et al., 2024d) | 24.3 | 87.5 | 88.8 | 48.7 | 74.2 | 37.1 | 76.1 | 33.1 | - | - | - | - | - | 52.6 | 62.7 | 32.8 | 38.9 | 31.6 |
| LLaVA-N-Inter-14B (Li et al., 2024d) | 24.5 | 95.0 | 91.1 | 59.9 | 76.7 | 40.5 | 78.6 | 33.3 | 70.6 | 52.2 | 34.5 | 62.0 | 76.7 | 52.1 | 66.4 | 33.4 | 40.7 | 32.7 |
| *LLaVA-OV-0.5B (SI)* | *15.6* | *44.8* | *56.1* | *30.0* | *45.8* | *8.5* | *36.7* | *7.6* | *22.1* | *22.1* | *16.9* | *25.5* | *8.2* | *37.9* | *38.2* | *20.9* | *22.7* | *26.7* |
| LLaVA-OV-0.5B | 17.1 | 48.7 | 63.4 | 35.4 | 48.8 | 36.4 | 65.0 | 29.8 | 60.0 | 48.0 | 29.4 | 62.2 | 70.5 | 52.1 | 39.6 | 60.0 | 25.5 | 29.1 |
| *LLaVA-OV-7B (SI)* | *20.5* | *60.3* | *75.9* | *24.6* | *56.0* | *7.9* | *52.8* | *8.4* | *24.5* | *29.9* | *22.1* | *32.0* | *70.8* | *45.6* | *54.2* | *26.3* | *32.7* | *30.0* |
| LLaVA-OV-7B | 22.2 | 90.2 | 89.4 | 53.3 | 74.5 | 39.2 | 80.1 | 31.7 | 62.8 | 52.6 | 30.1 | 61.0 | 79.8 | 48.2 | 64.2 | 67.6 | 41.8 | 79.1 |
| *LLaVA-OV-72B (SI)* | *22.1* | *61.2* | *78.9* | *44.2* | *61.5* | *15.6* | *67.9* | *12.1* | *30.8* | *25.4* | *21.9* | *43.5* | *75.5* | *46.0* | *56.8* | *58.6* | *33.2* | *65.8* |
| LLaVA-OV-72B | 22.5 | 95.3 | 93.8 | 63.4 | 83.2 | 43.3 | 83.7 | 34.5 | 63.2 | 53.3 | 35.8 | 66.3 | 78.8 | 55.4 | 77.6 | 91.6 | 54.8 | 94.9 |

Table 4: LLaVA-OneVision performance on multi-image benchmarks with all results reported in accuracy. [†] denotes the LLaVA-NeXT-Vicuna-7B (2024-01). We use IEI for Image Edit Instruction, MI-VQA for Multi-image VQA, NLVR2 for Natural Language for Visual Reasoning, SDiff for Spot the Difference, VST for Visual Story Telling, TR-VQA for Text-rich VQA. For MathVerse and SciVerse, we report the accuracy on their multi-image splits.

| Model | ActNet-QA | EgoSchema | MLVU | MVBench | NextQA | PercepTest | SeedBench | VideoChatGPT | VideoDC | VideoMME | L-VideoBench |
|---|---|---|---|---|---|---|---|---|---|---|---|
| | test | test | m-avg | test | mc | val | video | test | test | wo/w-subs | val |
| GPT-4V (OpenAI, 2023) | 57.0 | - | 49.2 | 43.5 | - | - | 60.5 | 4.06 | 4.00 | 59.9/63.3 | 61.3 |
| GPT-4o (OpenAI, 2024) | - | - | 64.6 | - | - | - | - | - | - | 71.9/77.2 | 66.7 |
| Gemini-1.5-Flash (Team et al., 2023) | 55.3 | 65.7 | - | - | - | - | - | - | - | 70.3/75.0 | 61.6 |
| Gemini-1.5-Pro (Team et al., 2023) | 57.5 | 72.2 | - | - | - | - | - | - | - | 75.0/81.3 | 64.0 |
| VILA-40B (Lin et al., 2024) | 58.0 | 58.0 | - | - | 67.9 | 54.0 | - | 3.36 | 3.37 | 60.1/61.1 | - |
| PLLaVA-34B (Xu et al., 2024a) | 60.9 | - | - | 58.1 | - | - | - | 3.48 | - | - | - |
| LLaVA-N-Video-34B (Zhang et al., 2024h) | 58.8 | 49.3 | - | - | 70.2 | 51.6 | - | 3.34 | 3.48 | 52.0/54.9 | 50.5 |
| LongVA-7B (Zhang et al., 2024d) | 50.0 | - | 56.3 | - | 68.3 | - | - | 3.20 | 3.14 | 52.6/54.3 | - |
| IXC-2.5-7B (Zhang et al., 2024c) | 52.8 | - | 37.3 | 69.1 | 71.0 | 34.4 | - | 3.46 | 3.73 | 55.8/58.8 | - |
| LLaVA-N-Video-32B (Zhang et al., 2024h) | 54.3 | 60.9 | 65.5 | - | 77.3 | 59.4 | - | 3.59 | 3.84 | 60.2/63.0 | - |
| *LLaVA-OV-0.5B (SI)* | *49.0* | *33.1* | *47.9* | *43.3* | *53.6* | *48.6* | *43.4* | *3.08* | *3.51* | *41.7/40.4* | *41.9* |
| LLaVA-OV-0.5B | 50.5 | 26.8 | 50.3 | 45.5 | 57.2 | 49.2 | 44.2 | 3.12 | 3.55 | 44.0/43.5 | 45.8 |
| *LLaVA-OV-7B (SI)* | *55.1* | *52.9* | *60.2* | *51.2* | *61.6* | *54.9* | *51.1* | *3.54* | *3.51* | *55.0/59.1* | *54.3* |
| LLaVA-OV-7B | 56.6 | 60.1 | 64.7 | 56.7 | 79.4 | 57.1 | 56.9 | 3.51 | 3.75 | 58.2/61.5 | 56.4 |
| *LLaVA-OV-72B (SI)* | *62.1* | *58.6* | *60.9* | *57.1* | *67.2* | *62.3* | *60.9* | *3.55* | *3.66* | *64.8/66.9* | *58.3* |
| LLaVA-OV-72B | 62.3 | 62.0 | 68.0 | 59.4 | 80.2 | 66.9 | 62.1 | 3.62 | 3.60 | 66.2/69.5 | 61.3 |

Table 5: LLaVA-OneVision performance on video benchmarks. We report the score out of 5 for VideoDC, VideoChatGPT while other results are reported in accuracy. All results are reported as 0-shot accuracy.

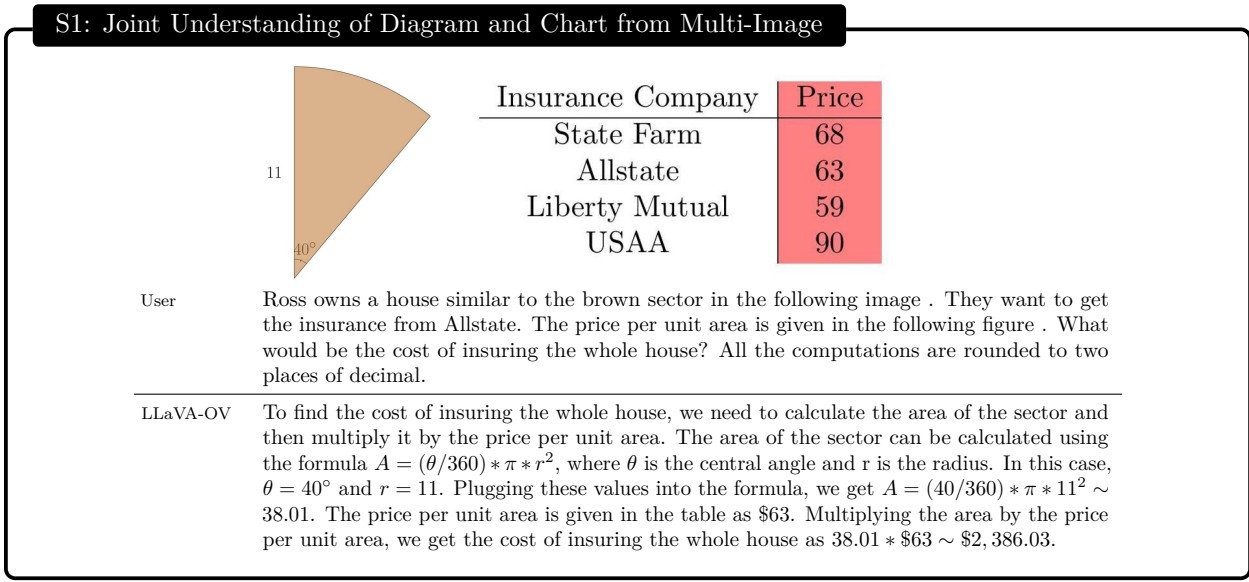

> **S1: Joint Understanding of Diagram and Chart from Multi-Image**
>
> | Insurance Company | Price |
> |:---:|:---:|
> | State Farm | 68 |
> | Allstate | 63 |
> | Liberty Mutual | 59 |
> | USAA | 90 |
>
> **User**  Ross owns a house similar to the brown sector in the following image . They want to get the insurance from Allstate. The price per unit area is given in the following figure . What would be the cost of insuring the whole house? All the computations are rounded to two places of decimal.
>
> **LLaVA-OV**  To find the cost of insuring the whole house, we need to calculate the area of the sector and then multiply it by the price per unit area. The area of the sector can be calculated using the formula $A = (\theta/360) * \pi * r^2$, where $\theta$ is the central angle and r is the radius. In this case, $\theta = 40°$ and $r = 11$. Plugging these values into the formula, we get $A = (40/360) * \pi * 11^2 \sim$ 38.01. The price per unit area is given in the table as \$63. Multiplying the area by the price per unit area, we get the cost of insuring the whole house as $38.01 * \$63 \sim \$2,386.03$.

Table 6: LLaVA-OneVision transfers its ability to *understand diagram and table to multi-image scenarios*, interpreting multiple images in a coherent manner.

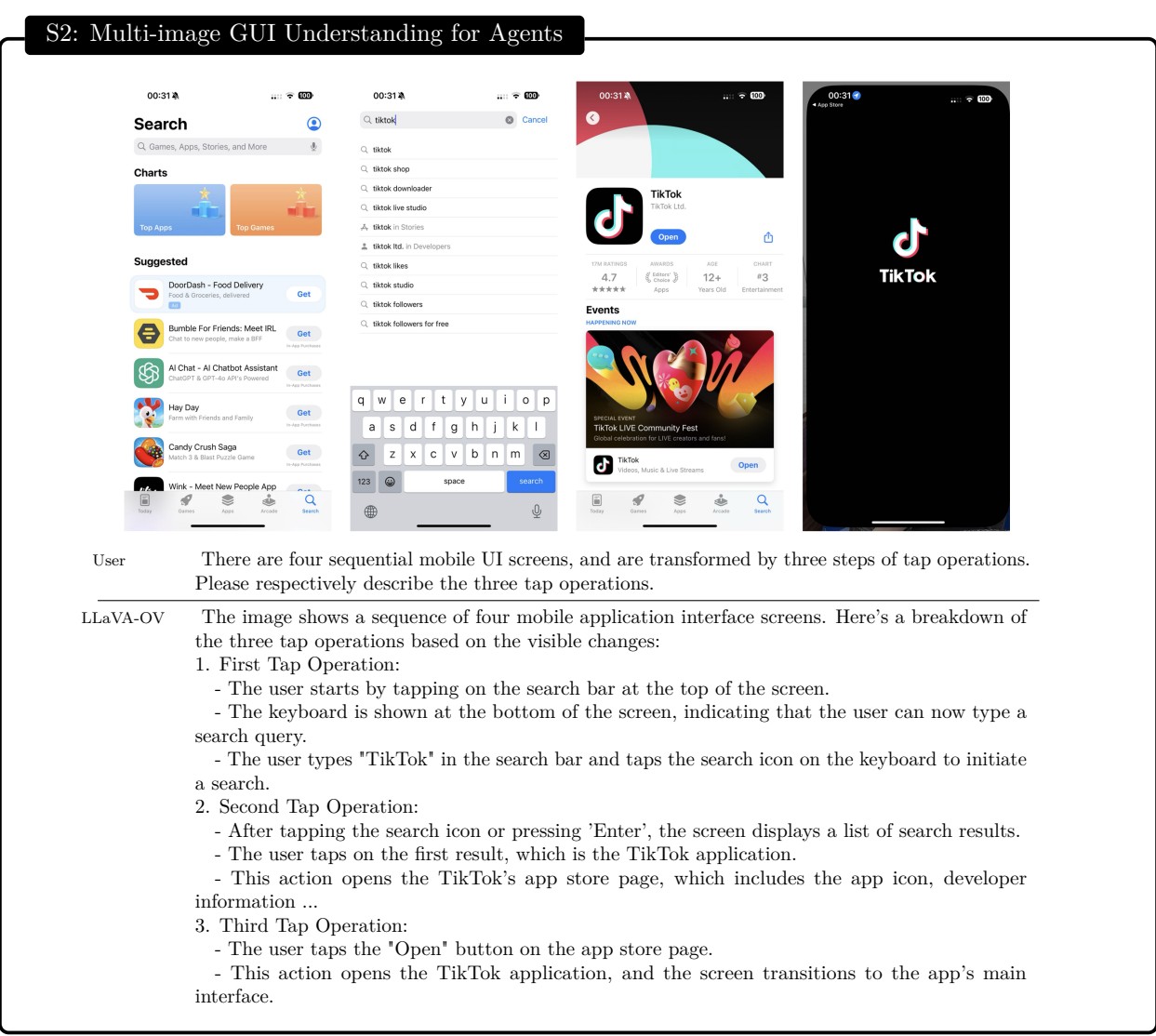

**S2: Multi-image GUI Understanding for Agents**

| User | There are four sequential mobile UI screens, and are transformed by three steps of tap operations. Please respectively describe the three tap operations. |
|------|--------|
| LLaVA-OV | The image shows a sequence of four mobile application interface screens. Here's a breakdown of the three tap operations based on the visible changes: |

1. First Tap Operation:
   - The user starts by tapping on the search bar at the top of the screen.
   - The keyboard is shown at the bottom of the screen, indicating that the user can now type a search query.
   - The user types "TikTok" in the search bar and taps the search icon on the keyboard to initiate a search.
2. Second Tap Operation:
   - After tapping the search icon or pressing 'Enter', the screen displays a list of search results.
   - The user taps on the first result, which is the TikTok application.
   - This action opens the TikTok's app store page, which includes the app icon, developer information ...
3. Third Tap Operation:
   - The user taps the "Open" button on the app store page.
   - This action opens the TikTok application, and the screen transitions to the app's main interface.

Table 7: LLaVA-OneVision *plays the role of agent.* It recognizes multiple screenshots on the iPhone and take action to interact with the iPhone, providing operation instructions for automating tasks.

---

**S3: Set-of-mark Prompting**

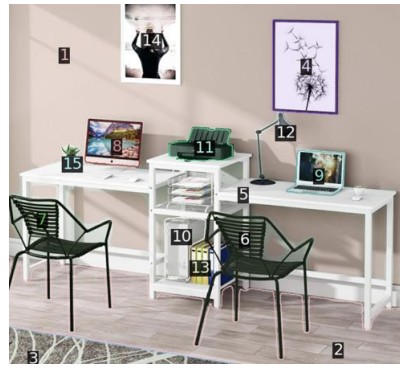

| User | Here's an image with objects labeled in numbers. Please describe the objects that are marked as 4, 5 and 7. |
|---|---|
| LLaVA-OV | The image shows a room with various items marked with numbers. Here's the description for the items marked as 4, 5, and 7:
The item marked as 4 is a framed picture hanging on the wall. It features a black and white illustration of a dandelion with seeds blown away, creating a sense of movement.
The item marked as 5 is a white bookshelf with multiple shelves. It appears to be a storage unit that is part of the room's furniture. It is located between the two desks.
The item marked as 7 is a black chair with a modern design. It has a metal frame and a woven seat, suggesting a contemporary style. This chair is positioned in front of one of the desks. |

---

Table 8: LLaVA-OneVision exhibits *excellent set-of-mark prompting capabilities*, ie, referring to marks when answering questions. This example demonstrates that describing specific objects based on numerical labels within an image highlights its comprehension skills in handling fine-grained visual content.

---

**S4: Image-to-Video Editing Instruction**

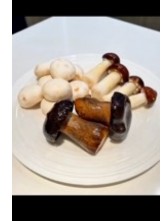 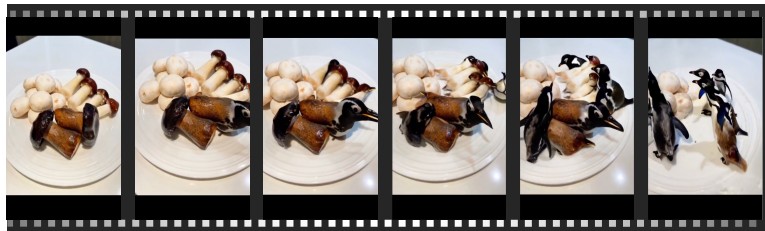

| User | Please provide the video generation instruction that can change the source image to the target video. |
|---|---|
| LLaVA-OV | To generate a video from the provided image, you would need to create a sequence of images that depict the progression of the mushrooms being transformed into penguins. Here's a step-by-step guide on how to create this video:
1. Start with the initial image of the mushrooms on a plate.
2. In the next image, begin to change the appearance of the mushrooms to resemble penguins. Start with the smallest mushrooms and gradually move to the larger ones.
3. Continue to transform the mushrooms into penguins, adding details such as eyes, beaks, and flippers.
4. As the video progresses, more mushrooms should be transformed, and the penguins should appear to interact with each other and the mushrooms.
5. The final images should show a group of penguins interacting with the mushrooms, creating a playful and whimsical scene. Remember to maintain a consistent style and color palette throughout the video to ensure a cohesive look. The transformation should be smooth and realistic, making it seem as though the mushrooms are naturally turning into penguins. |

---

Table 9: LLaVA-OneVision learns to *generate detailed video creation prompts based on a static image.* This capability is generalized to videos from the image-to-image language editing generation.

**S5: Video-to-Video Difference**

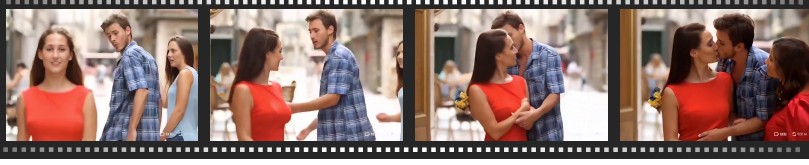

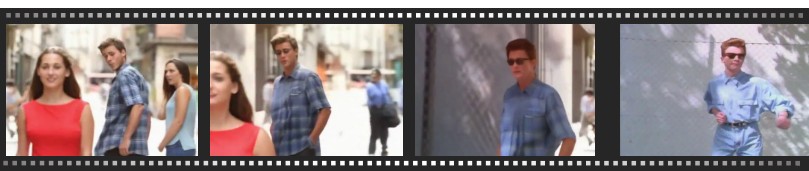

| User | Here are two videos, please provide the analysis on their differences. |
|---|---|
| LLaVA-OV | The first video shows a sequence of images where a man in a blue plaid shirt approaches a woman in a red dress, engages in a conversation, and eventually gives her a kiss on the cheek. The woman in the red dress is looking at the man with a slight smile, and another woman in a light blue top is observing the scene. The setting appears to be a street with buildings in the background. |
|  | The second video shows the same man in the blue plaid shirt walking away from the woman in the red dress, who is now looking in the opposite direction. The woman in the light blue top is no longer visible in this sequence. The man is walking down the street, passing by the buildings and other pedestrians. The overall mood of the video changes from the intimate and personal interaction in the first video to a more solitary and neutral scene in the second video. |

Table 10: LLaVA-OneVision learns to *analyze differences between videos* with the same starting frame but different endings.

| | |
|---|---|
| **S5: Video-to-Video Difference** | |
| User | What are the differences between the two videos? |
| LLaVA-OV | The first video features a caterpillar with green and black stripes on its body, resting on a leaf. The caterpillar has white, bristly hair-like structures along its back. The background is blurred, but it appears to be a garden or a natural setting with green foliage and some sunlight filtering through. The second video shows a butterfly with blue and black wings, resting on a leaf. The butterfly has white spots on its wings and an orange patch near the base of the wings. The background is also blurred but suggests a similar garden or natural setting with greenery. The differences between the two videos are: |
| | 1. The first video shows a caterpillar, while the second video shows a butterfly. |
| | 2. The caterpillar has green and black stripes, while the butterfly has blue and black wings with white spots. |
| | 3. The caterpillar has white, bristly hair-like structures along its back, which are not present on the butterfly. |
| | 4. The butterfly has a more delicate and transparent appearance compared to the fuzzy texture of the caterpillar. |
| | 5. The wings of the butterfly are spread out, while the caterpillar is in a curled position. |
| | 6. The butterfly has long antennae, which are not visible in the caterpillar's frame. |

Table 11: LLaVA-OneVision learns to *analyze differences between videos* with similar backgrounds but different foreground objects.

## S6: Multi-camera Video Understanding in Self-driving

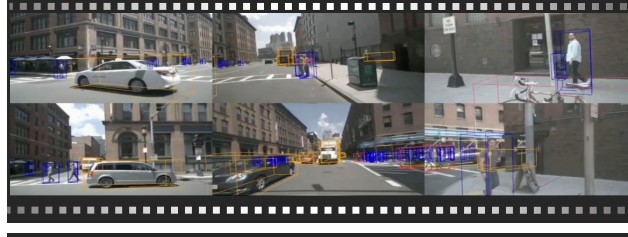

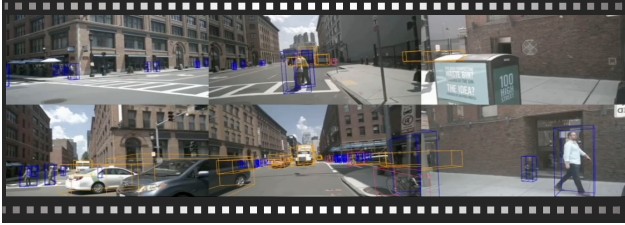

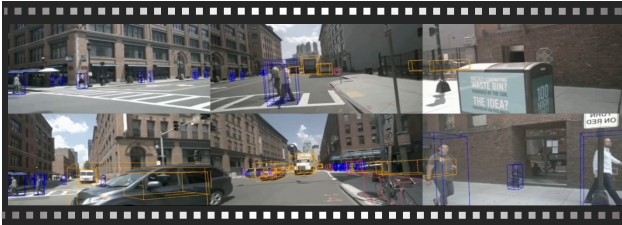

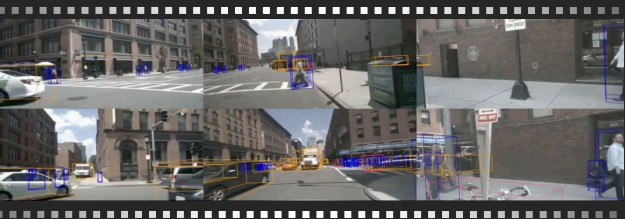

**User** Here is a video showing four camera views of a driving car. The top two show front views, and the bottom two show back views. First describe the video of each view in detail, and then plan the next move of the ego car.

**LLaVA-OV** The video consists of six different camera views of a driving car, arranged in two columns with three rows. Each view captures the car from a different angle, showing both front and back perspectives.
In the front views:
   1. The first image shows the car approaching an intersection with a pedestrian crossing. There are cars ahead, and the pedestrian is visible on the sidewalk.
   2. The second image shows the car at the intersection, with the pedestrian crossing and cars ahead. The pedestrian is now closer to the car.
   3. The third image shows a pedestrian visible on the sidewalk.
In the back views:
   1. The first image shows the car approaching the intersection with pedestrians crossing and a black car aside.
   2. The second image shows the car at the intersection, with several cars behind.
   3. The third image shows the car passing through the intersection, with the pedestrian visible on the sidewalk.
The next move of the ego car would be to continue driving forward, following the road and traffic rules, and ensuring the pedestrian's safety as they cross the street.The next move of the ego car would be to continue driving forward, following the road and traffic rules, and ensuring the pedestrian's safety as they cross the street.

Table 12: LLaVA-OneVision *analyzes and interprets multi-camera video footage in self-driving contexts.*

**S7: Composed Sub-video Understanding**

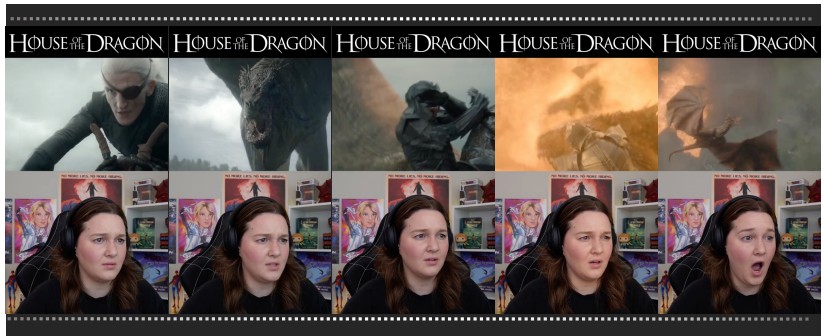

| User | Please describe the video's content and layout in detail. |
|------|-----------------------------------------------------------|
| LLaVA-OV | The video showcases a series of frames featuring a central, obscured foreground view of a room, presumably a game room or office setup. The room is adorned with several posters on the wall, a model of a humanoid figure wearing armor, and various other items that suggest a space dedicated to hobbies or gaming. In the foreground, there is a person, likely a woman, who is wearing headphones and has a neutral to slightly surprised facial expression, indicating that she may be reacting to something off-screen. |

The backdrop throughout the video is a dynamic, digital illustration from the television show "Game of Thrones," specifically from the series "House of the Dragon." This series is a prequel to "Game of Thrones" and centers around the House Targaryen, a prominent family within the show's fictional world. The images in question feature characters from the series, including members of House Targaryen. Some characters are seen in battle attire, riding dragons, and engaged in combat scenarios. The scenes are detailed and vivid, depicting intense moments from the show, such as the White Walkers and dragon battles.

The video starts with the person in the foreground wearing headphones and a black shirt. Behind her, the consistent backdrop from "House of the Dragon" unfolds, focusing on characters from the series. It is evident that the series elements are meant to evoke a sense of drama and tension that is central to the narrative of "House of the Dragon."

As the video progresses, the scenes transition through various moments from the series. The characters are shown in different scenarios, such as riding dragons, engaging in combat, and experiencing intense moments. The visuals are rich with color and detail, highlighting the epic scale and dramatic tone of the series. The consistent presence of the game room setup in the foreground provides a sense of continuity throughout the video.

The video captures the essence of "House of the Dragon" by using dramatic and action-packed scenes from the series. The person in the foreground remains a constant, adding an element of personal reaction to the intense moments presented in the background.

In summary, the video juxtaposes a calm, possibly surprised individual in a game room setup with dynamic and intense scenes from "House of the Dragon," creating a contrast between the peaceful foreground and the dramatic background

Table 13: LLaVA-OneVision learns to *understand and describe composed sub-videos in detail.*

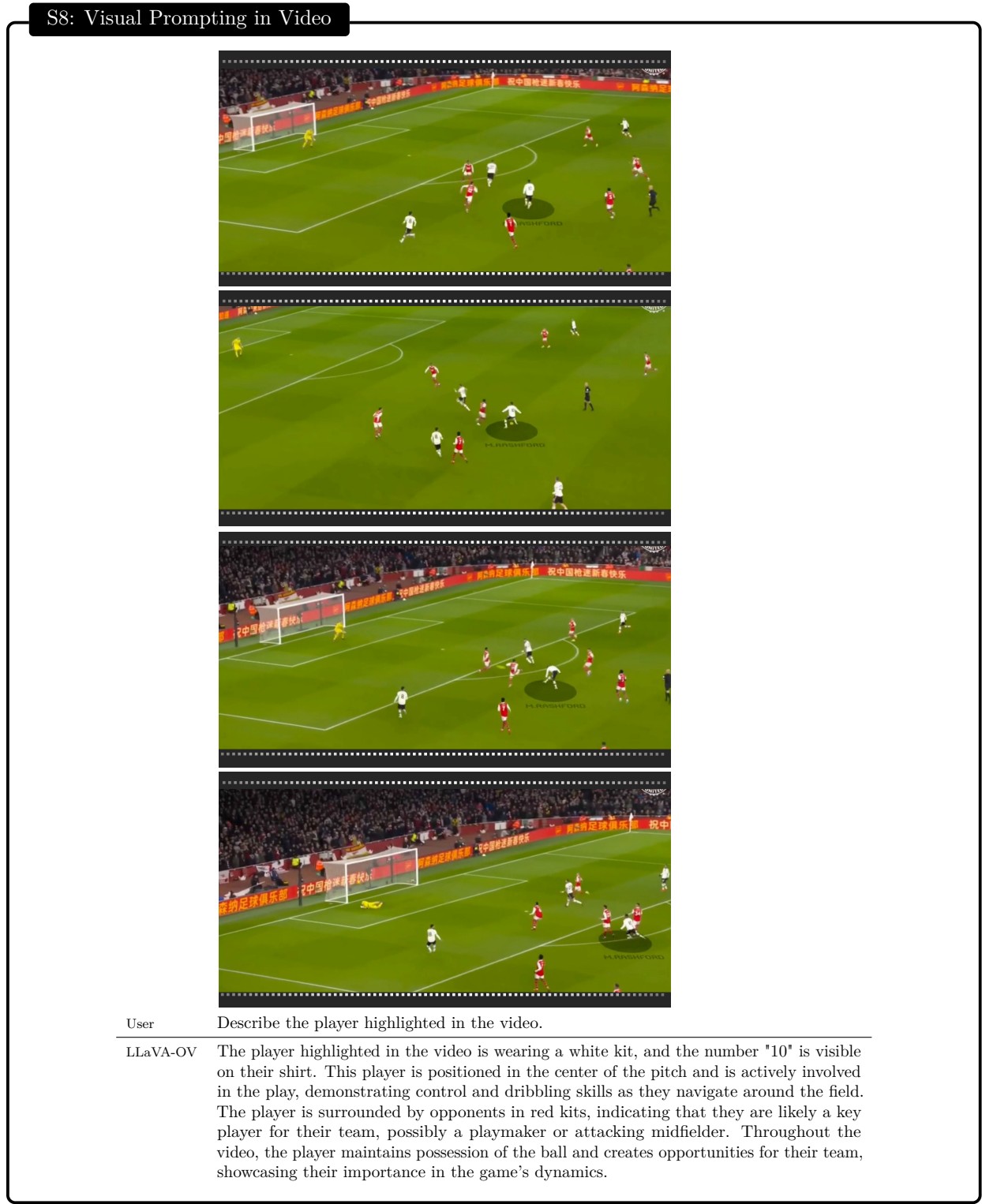

| | |
|---|---|
| User | Describe the player highlighted in the video. |
| LLaVA-OV | The player highlighted in the video is wearing a white kit, and the number "10" is visible on their shirt. This player is positioned in the center of the pitch and is actively involved in the play, demonstrating control and dribbling skills as they navigate around the field. The player is surrounded by opponents in red kits, indicating that they are likely a key player for their team, possibly a playmaker or attacking midfielder. Throughout the video, the player maintains possession of the ball and creates opportunities for their team, showcasing their importance in the game's dynamics. |

Table 14: LLaVA-OneVision learns to *provide detailed descriptions of highlighted subjects in video content.*

## S9: Referring Image in Video Understanding

| | |
|---|---|
| User | There are 2 images. Does the person in image 1 appear in image 2? If yes, what is he doing ? |
| LLaVA-OV | Yes, the person from image 1 appears in image 2. He is playing soccer with two other individuals in a residential outdoor area. |
| User | Who is the man in image 1? |
| LLaVA-OV | The man is Lionel Messi. |

| | |
|---|---|
| User | Here is an image and a video. Does the man shown in the image also appear in the video? If yes, describe his behavior in detail. |
| LLaVA-OV | Yes, the man shown in the image appears in the video. He is seen engaging in a soccer activity in a grassy outdoor area. He is dressed in a white shirt, dark pants, and white shoes. Throughout the video, he is seen standing near a yellow and black soccer ball, and at various points, he is either preparing to kick the ball or has just kicked it. His actions suggest he is actively participating in the game, and he appears to be in motion, indicating movement and activity. |

| | |
|---|---|
| User | Here is a video and an image. Does the man shown in the image also appear in the video? Explain it. |
| LLaVA-OV | The man shown in the image does not appear in the video. The image shows a person in a red sports jersey with the number 7, while the video features individuals playing with a soccer ball in a grassy outdoor area. The clothing, background, and activity are different between the image and the video, indicating that they are separate and not related. |
| User | Who is the man in the image? |
| LLaVA-OV | The man is Cristiano Ronaldo. |

Table 15: LLaVA-OneVision's capability in *referring image and video understanding.* It accurately identifies the same individual in two images in the first instance. It identifies the same individual in both the image and the video in the second instance and correctly concludes the absence of the individual in the third instance, indicating its understanding capability to relate visual query in both image and video understanding.

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

# A    Implmenetation Details

## A.1    Token Strategy for Mixed-Modality Data

We provide a detailed explanation of our token strategy for handling mixed-modality data within LLaVA-OneVision's architecture, which is illustrated in Figure 3.

*For single-image data,* we employ the *AnyResMax-9* strategy, as previously outlined in blog Li et al. (2024a). Using SO400M Zhai et al. (2023) as the Vision Encoder, each input image (or grid) is processed into 729 visual tokens. Consequently, the maximum number of visual tokens for a single image is $729 \times (1 + 9)$, where $1 \times 729$ represents the base tokens and $9 \times 729$ accounts for the grid tokens.

*For multi-image data,* we utilize a simple padding strategy. Each image is first resized to fit within a 384x384 frame by zero-padding, as required by SO400M, while maintaining the aspect ratio. After processing through the vision encoder, the zero-padding is removed from the tokens. Our training data includes up to 12 images per instance, resulting in a maximum of $12 \times 729$ multi-image tokens.

*For video data,* we adopt a strategy similar to LLaVA-NeXT-Video Zhang et al. (2024h). Each frame is processed through the vision encoder and then subjected to $2 \times 2$ bilinear interpolation, resulting in 196 tokens per frame. We sample up to 32 frames per video, leading to a maximum of $32 \times 196$ video tokens.

As shown in Figure 3, the maximum number of tokens across different modalities is approximately equal. This design strategy aims to balance the data from various modalities, ensuring more equitable representation that is transferable from the perspective of the language model. For instance, a high-resolution image can be interpreted as a composition of multiple images, and multiple images can be understood as a shorter video.

## A.2    Language Templates and Special Tokens

We utilize the Qwen-2 series Yang et al. (2024) language models with the template as OpenAI's ChatML[1]. During training, we adopt `<image>` as the marker for image tokens, following previous LLaVA models. This image special token is represented as $-200$ in the input index after tokenization. For multi-image scenarios, we use multiple `<image>` interleaved with text to denote the positions of the images. For video scenarios, we place a single `<image>` at the beginning to indicate the inclusion of a video.

One more aspect related to the handling of image tokens is ensuring that there are no extra `<image>` in the data. For instance, in some code writing tasks, there could be `<image>...</image>` related to HTML code. To avoid potential misunderstandings, we manually removed around 10 such samples from the Magpie Xu et al. (2024b) and Screen2Words Wang et al. (2021) datasets.

# B    Evaluation Steers Development

## B.1    Post-Evaluation as a Development Tool

With the help of our comprehensive evaluation toolkit, LMMs-Eval Zhang et al. (2024b), we conduct post-evaluations on a selected set of benchmarks after each training experiment concludes.

Our preference for selecting benchmarks is based on whether the targeted scenarios are sufficiently important and specific. These evaluations should not be too resource-intensive, meaning the benchmarks should not contain too many items, take too long to evaluate, or consume a large number of GPT-4V tokens (when using it as the judge model).

In our development, we evaluate on AI2D Kembhavi et al. (2016a), ChartQA Masry et al. (2022), DocVQA Mathew et al. (2021), and InfoVQA Mathew et al. (2022) to examine the model's fine-grained understanding of tables, charts, and diagrams, as well as MME Fu et al. (2024a) for formatting control, since it requires only Yes or No answers. We also include MMBench-Dev Liu et al. (2023e) and MMMU-Val Yue

---

[1]OpenAI Release v0.28.0/chatml.md

et al. (2024) for multi-discipline evaluation. Quickly obtaining evaluation results on these benchmarks will guide our next steps in model development and data curation.

## B.2 Improving Model Performance on Key Scenarios

During our development process, we gradually recognized the significance of using static evaluation benchmarks as perfprmance indicators. Our primary goal at this stage is not to overfit the model to certain datasets to achieve exceptionally high performance. Instead, we benchmark our models against GPT-4V's performance to set our target thresholds (e.g., initially 80%, gradually increasing to 95%-100%). Once the model meets the score requirements in static evaluations, it indicates that the model has sufficient capabilities in the selected scenarios. Furthermore, we cannot blindly pursue results on benchmarks, as even the test data for AI2D may have certain issues [2].

Ultimately, our focus is on optimizing the model's visual chat and reasoning capabilities. In this stage, we monitored the model's performance on benchmarks such as MathVista Lu et al. (2023a), LLaVA-Wilder Li et al. (2024b), MM-LiveBench Zhou et al. (2024b), and Vibe-Eval Padlewski et al. (2024a). These benchmarks require the model to engage in visual dialogue with challenging questions, and demand a diverse skill set with extensive world knowledge. This helps us create a model with strong generalization capabilities in real-world scenarios.

## B.3 Evaluation Task Information

In this section, we provide information on all the tasks used during the evaluation. Specifically, we use the default `post_prompt` and `pre_prompt` from the `LMMs-Eval` framework. These prompts are consistent with the evaluation of our previous LLaVA-NeXT Li et al. (2024b); Zhang et al. (2024h); Li et al. (2024d). The table below details the specific tasks used in `LMMs-Eval` and their corresponding task names.

---

**Tasks Information**
- **Single-image:**
  - ai2d, chartqa, docvqa__val, infovqa__val, mme, realworldqa, mathvista__testmini, llava__in__the__wild, mmvet, mmbench__en__dev, ocrbench, mmmu, llava__wilder__small, vibe__eval, wildvision__0617, live__bench__2406, mathverse__testmini__vision, seedbench, scienceqa__img, mmstar, dc100__en
- **Videos:**
  - activitynetqa, videochatgpt, nextqa__mc__test, egoschema, video__dc499, videmme, videomme__w__subtitle, perceptiontest__val__mc, mlvu, mvbench, longvideobench__val__v
- **Multi-image:**
  - llava__interleave__bench, muirbench

---

By referring to the task names listed here, the audience can directly retrieve the generation arguments and specific prompt information. For instance, the details for `tasks=ai2d` are available at lmms-eval/ai2d. By following these settings, researchers can easily reproduce our results.

# C  Data Curation Roadmap of LLaVA-NeXT Series

In this section, we provide the in-depth experience and roadmap of data curation in the LLaVA-NeXT series. To achieve strong multimodal performance, we need to collect and curate high-quality data from various sources, which is crucial for the model's generalization capabilities.

## C.1  Single-Image Data Curation

As the primary data source, our principle for single-image data has always been that quality outweighs quantity. Given limited resources, we strive to use high-quality data to maximize the performance.

---

[2]Discussion on AI2D Evaluation

The first version of the LLaVA-NeXT models (LLaVA-NeXT-Vicuna-7B/13B, Mistral-7B, Hermes-Yi-34B), comprising 760K data samples Liu et al. (2024b), includes 665K samples from LLaVA-1.5 Liu et al. (2024a), 3,247 samples from AI2D Kembhavi et al. (2016b), 18,317 samples from ChartQA Masry et al. (2022), 10,194 samples from DocVQA Mathew et al. (2021), 20,000 samples from DVQA Kafle et al. (2018), 40,093 samples from SynthDOG-EN Kim et al. (2022), and 15,131 samples from user requests on LLaVA's demo, re-annotated with GPT-4V. In the subsequent iteration, we added 20,000 samples from COCO Caption Lin et al. (2015), forming a new 790K version. This 790K dataset supported the second release of LLaVA-NeXT models (LLaVA-NeXT-LLaMA3-8B, LLaVA-NeXT-Qwen-72B, LLaVA-NeXT-Qwen-110B).

In subsequent collections, we accumulated open-sourced datasets from the Internet and referred to the dataset collection processes of other advanced LMMs, such as Qwen-VL Bai et al. (2023b), DeepSeek-VL Lu et al. (2024), Intern-VL Chen et al. (2023b), Vision-Flan Xu et al. (2024d), UReader Ye et al. (2023), Idefics-2 (Cauldron) Laurençon et al. (2024), and Cambrian. During the data iteration process, we strictly adhered to the initial LLaVA-1.5 strategy. For each dataset, we manually inspected and ensured its quality and QA format. We also designed specific formatting prompts to make data from different sources compatible with each other, thus avoiding conflicts.

Some data sources, such as AI2D and ChartQA, appear in different dataset collections and may be duplicated. Since Cauldron includes special formatting prompts, its data is not straightforward to re-format. Therefore, we prioritize using data from other collections that are closer to the raw format. For the Cambrian dataset, we only selected a subset of the GPT-4o re-annotated data. We also collected math-related data from the MathV and MAVIS datasets.

For the pure language data, we replaced the ShareGPT ShareGPT (2023) text data that LLaVA has been using since version 1.5. Given that our largest Qwen2-72B model has achieved performance levels close to latest GPT-4 model in language tasks, we need to use higher quality language data to maintain or further enhance its language capabilities. To achieve this, we sourced the highest quality language SFT data available, the Magpie-Pro dataset Xu et al. (2024b).

After undergoing the aforementioned process, we have obtained approximately 4 million raw SFT data samples, ensuring their quality and accuracy. Additionally, we utilized Azure's OpenAI GPT-4V and GPT-4o services to re-annotate our data, focusing on scenarios that were not adequately covered by the original data but are crucial. These scenarios include:

**(1) Detailed Descriptions on Charts and Diagrams:** For this scenario, we used images from the AI2D and InfoVQA training sets and employed GPT-4V to provide detailed descriptions of the images, resulting in 4,874 detailed descriptions for AI2D and 1,992 samples for InfoVQA.

**(2) Chinese Language:** We used images from the LLaVA-158K dataset and employed GPT-4o to provide detailed descriptions in Chinese, resulting in a total of 91,466 samples.

**(3) Multi-turn Dialogue:** Also with the LLaVA-158K dataset, we employed GPT-4o to create long dialogues with an average of more than 3 turns per conversation, obtaining a total of 26,048 samples.

When resources permit, we recommend a data validation process we used in early stage data sourcing. We extract approximately 100K samples from each newly added data source or collection (if the selected data source can form a collection) and add them to the 790K version of the dataset. We validate newly added data under the SO400M-Qwen-1.5-0.5B experimental setting. If the addition of new data results in a performance decline compared to the baseline, we conduct further manual inspections of the data and adjust the formatting prompt accordingly. This step requires abundant resources and must be carried out by highly professional researchers, as it cannot be substituted with average human annotators.

During the collection process, we manually labeled the datasets with two tags: {General, Language, Math-/Reasoning, General OCR, Doc/Chart/Screen} and {Fixed-form, Free-form}. Based on these tags, we formed the final distribution of 3.2 million single-image data samples.

Starting with the initial distribution, we gradually increased the amount of free-form (most of them are GPT-4V/o annotated) data and observed the model's performance on various benchmarks and try to balance among them. These benchmarks include academic datasets, such as AI2D Kembhavi et al. (2016a), MME Fu

et al. (2024a), MMMU Yue et al. (2024), MathVista Lu et al. (2023a), and visual chat datasets, such as LLaVA-Wilder Li et al. (2024b), and Vibe-Eval Padlewski et al. (2024a). Ultimately, we gradually established an optimal data distribution for single-image tasks under the 7B setting.

## C.2 OneVision Data Curation

In addition to single-image data, we incorporate multi-image and video datasets to support a wider scope of visual scenarios. We aim to balance the capability among different data modalities, and achieve an overall superior performance with one framework as LLaVA-OneVision.

For multi-image data, we adopt the diverse interleaved multimodal tasks within M4-Instruct dataset from LLaVA-NeXT-Interleave Li et al. (2024d). This dataset mainly comprises general multi-image tasks, such as spotting the difference, visual story telling, image editing instruction generation, interleaved multi-image dialogue, multi-image puzzle, low-level multi-image assessment, etc. Besides, we also utilize the multi-view datasets in M4-Instruct to indicate spatial information in the 3D world, including embodied VQA (dialogue and planning) and 3D scene VQA (captioning and grounding).

For video data, we first integrate the multi-frame data from M4-Instruct, including NExT-QA Xiao et al. (2021) and ShareGPT4Video Chen et al. (2024b). Then, to enable more detailed temporal cues, we select several datasets commonly used in recent academic research for re-annotation, including Charades Sigurdsson et al. (2016), ActivityNet Yu et al. (2019), YouCook2 Zhou et al. (2017), and Ego4D Grauman et al. (2022). Initially, we annotated captions. Following ShareGPT-4o Laboratory (2023), we sampled video frames at 1 frame per second (FPS) and used the pre-defined instructions to prompt GPT-4o for generating video captions. Additionally, following LLaVA-Hound Zhang et al. (2024g), we developed open-ended question-answering pairs and their corresponding multiple-choice versions using the captions created by GPT-4o. We also employed GPT-4o to generate question-answer pairs, obtaining high-quality video data for OneVision training.

## C.3 Detailed Dataset Statistics

We primarily use tables to present the statistical information of all datasets utilized in both the Single-Image and OneVision stages. The information includes the dataset category, dataset name, number of samples, and prompt type. The dataset statistics are summarized in Table 16.

| Dataset | # Samples | Prompt ID | Dataset | # Samples | Prompt ID |
|---|---|---|---|---|---|
| **General (1.14M, 36.1%)** | | | | | |
| AOKVQA Schwenk et al. (2022) | 66160 | 1 | Cambrian (filtered) Tong et al. (2024) | 83131 | - |
| CLEVR Johnson et al. (2017) | 700 | 1 | COCO Caption Lin et al. (2015) | 20000 | 9 |
| Hateful Memes Kiela et al. (2020) | 8500 | 1 | IconQA Lu et al. (2021c) | 2494 | 5 |
| Image Textualization Pi et al. (2024) | 99583 | 11 | LLaVA-158K Liu et al. (2023c) | 158000 | - |
| LLaVA-Wild (train) Liu et al. (2023c) | 54517 | - | LLaVAR Zhang et al. (2023b) | 20000 | - |
| OKVQA Marino et al. (2019) | 8998 | 1 | RefCOCO Yu et al. (2016) | 50586 | 7,8 |
| ScienceQA Lu et al. (2022) | 4976 | 5 | ShareGPT4O ShareGPT (2023) | 57289 | 11 |
| ShareGPT4V ShareGPT (2023) | 92025 | 11 | ST-VQA Biten et al. (2019) | 17247 | 1 |
| TallyQA Acharya et al. (2019) | 9868 | 1 | Vision FLAN Xu et al. (2024d) | 186070 | - |
| Visual7W Zhu et al. (2016) | 14366 | 5 | VisText Tang et al. (2023) | 9969 | 15 |
| VizWiz Gurari et al. (2018) | 6614 | 2 | VQARAD Lau et al. (2018) | 313 | 1 |
| VQAv2 Antol et al. (2015) | 82783 | 1 | VSR Liu et al. (2023a) | 2157 | 3 |
| WebSight | 10000 | 18 | InterGPS Lu et al. (2021b) | 1280 | 5 |
| ALLaVA Instruct Chen et al. (2024a) | 70000 | - | | | |
| **Doc/Chart/Screen (20.6%, 647K)** | | | | | |
| AI2D (GPT4V Detailed Caption) | 4874 | 12 | AI2D (InternVL Chen et al. (2023b)) | 12413 | 4 |
| AI2D (Original) Kembhavi et al. (2016b) | 3247 | 5 | Chart2Text Obeid & Hoque (2020) | 26961 | 13 |
| ChartQA Masry et al. (2022) | 18317 | 1 | Diagram Image2Text | 300 | 17 |
| DocVQA Mathew et al. (2021) | 10194 | 1 | DVQA Kafle et al. (2018) | 20000 | 1 |
| FigureQA Kahou et al. (2018) | 1000 | 3 | HiTab Cheng et al. (2022) | 2500 | 1 |
| Infographic VQA Mathew et al. (2022) | 4404 | 1 | LRV Chart Liu et al. (2023b) | 1787 | - |
| RoBUT SQA | 8514 | - | RoBUT WikiSQL | 74989 | - |
| RoBUT WTQ | 38246 | 1 | Screen2Words Wang et al. (2021) | 15730 | 10 |
| TQA Kembhavi et al. (2017b) | 1365 | 5 | UReader Caption Ye et al. (2023) | 91439 | 9 |
| UReader IE Ye et al. (2023) | 17327 | 1 | UReader KG Ye et al. (2023) | 37550 | 14 |
| UReader QA Ye et al. (2023) | 252954 | 1 | VisualMRC Tanaka et al. (2021) | 3027 | - |
| **Math/Reasoning (20.1%,632K)** | | | | | |
| MAVIS Manual Collection Zhang et al. (2024f) | 87358 | 19 | MAVIS Data Engine Zhang et al. (2024f) | 100000 | 19 |
| CLEVR-Math Johnson et al. (2017) | 5290 | 2 | Geo170K Align Gao et al. (2023) | 60252 | - |
| Geo170K QA Gao et al. (2023) | 67833 | 19 | Geometry3K Lu et al. (2021b) | 2101 | 6 |
| GEOS Seo et al. (2015) | 508 | 6 | Geometry3K (MathV360K) Lu et al. (2021a) | 9734 | 6 |
| GeoMVerse (MathV360K) Kazemi et al. (2023) | 9303 | 20 | GeoQA+ (MathV360K) Chen et al. (2022b) | 17172 | 6 |
| MapQA (MathV360K) Chang et al. (2022) | 5235 | 1 | MathQA Amini et al. (2019) | 29837 | 19 |
| Super-CLEVR Li et al. (2023b) | 8652 | 2 | TabMWP Lu et al. (2023b) | 45184 | 2 |
| UniGeo Chen et al. (2022a) | 11959 | 6 | GQA Hudson & Manning (2019) | 72140 | 1 |
| LRV Normal Liu et al. (2023b) | 10500 | - | RAVEN Zhang et al. (2019) | 2100 | 3 |
| Visual Genome Krishna et al. (2016) | 86417 | 7,8 | | | |
| **General OCR (8.9%,281K)** | | | | | |
| ChromeWriting Wendler (2023) | 8835 | 21 | HME100K Yuan et al. (2022) | 74502 | 21 |
| IIIT5K Mishra et al. (2012) | 2000 | 22 | IAM Marti & Bunke (2002) | 5663 | 22 |
| K12 Printing | 12832 | 22 | OCR-VQA Mishra et al. (2019b) | 80000 | 1 |
| Rendered Text Wendler (2023) | 10000 | 22 | SynthDog-EN Kim et al. (2022) | 40093 | 16 |
| TextCaps Sidorov et al. (2020) | 21952 | 9 | TextOCR-GPT4V Carter (2024) | 25114 | 11 |
| **Pure Language (450K) (14.3%, 647K)** | | | | | |
| Magpie Pro Xu et al. (2024b) (L3 MT) | 149999 | - | Magpie Pro (L3 ST) | 150000 | - |
| Magpie Pro (Qwen2 ST) | 149996 | - | | | |

Table 16: The detailed statistics of Single-Image datasets used in LLaVA-OneVision. Prompt ID denotes the ID of Formatting Prompt which is corresponding to the ID in Table 18. - denotes no fromatting prompt is used.

| Dataset | # Samples | Prompt ID | Dataset | # Samples | Prompt ID |
|---------|-----------|-----------|---------|-----------|-----------|
| | | | Multi-image Scenarios | | |
| Spot-the-Diff Jhamtani & Berg-Kirkpatrick (2018b) | 10.8K | 20 | Birds-to-Words Forbes et al. (2019) | 14.3K | 21 |
| CLEVR-Change Park et al. (2019); Hosseinzadeh & Wang (2021) | 3.9K | 22 | HQ-Edit-Diff Hui et al. (2024) | 7.0K | 3 |
| MagicBrush-Diff Zhang et al. (2024a) | 6.7K | 4 | IEdit Tan et al. (2019) | 3.5K | 19 |
| AESOP Ravi et al. (2021) | 6.9K | 23 | FlintstonesSV Gupta et al. (2018) | 22.3K | 24 |
| PororoSV Li et al. (2019) | 12.3K | 25 | VIST Ting-Hao et al. (2016) | 26K | 4 |
| WebQA Chang et al. (2021) | 9.3K | 8 | TQA (MI) Kembhavi et al. (2017a) | 8.2K | 9 |
| OCR-VQA (MI) Mishra et al. (2019a) | 1.9K | 17 | DocVQA (MI) Mathew et al. (2021) | 1.9K | 18 |
| RAVEN Zhang et al. (2019) | 35K | 5 | MIT-StateCoherence Isola et al. (2015) | 1.9K | 11 |
| MIT-PropertyCoherence Isola et al. (2015) | 1.9K | 12 | RecipeQA ImageCoherence Yagcioglu et al. (2018) | 8.7K | 14 |
| VISION Bai et al. (2023a) | 9.9K | 13 | Multi-VQA Li et al. (2024e) | 5K | - |
| IconQA Lu et al. (2021c) | 34.6K | - | Co-Instruct Wu et al. (2024) | 50.0K | - |
| DreamSim Fu et al. (2023) | 15.9K | - | ImageCoDe Krojer et al. (2022) | 16.6K | - |
| nuScenes Caesar et al. (2020) | 9.8K | 10 | ScanQA Azuma et al. (2022b) | 25.6K | 7 |
| ALFRED Shridhar et al. (2020) | 22.6K | 16 | ContrastCaption Jiang et al. (2024) | 25.2K | - |
| VizWiz (MI) Gurari et al. (2018) | 4.9K | 6 | ScanNet Dai et al. (2017) | 49.9K | 7 |
| COMICS Dialogue Iyyer et al. (2017) | 5.9K | 15 | NLVR2 Suhr et al. (2019) | 86K | 26 |
| | | | Multi-frame (Video) Scenarios | | |
| NExT-QA Xiao et al. (2021) | 9.5K | 2 | ActivityNet Yu et al. (2019) | 6.5k | 1 |
| Ego-4D Grauman et al. (2022) | 0.8K | 2 | Charades Sigurdsson et al. (2016) | 23.6K | 1 |
| YouCook2 Zhou et al. (2017) | 41.9K | 2 | ShareGPT4Video Chen et al. (2024b) | 255K | - |

Table 17: The detailed statistics of Multi-Image and Video datasets used in LLaVA-OneVision. Prompt ID denotes the ID of Formatting Prompt corresponding to the ID in Table 19. - denotes no fromatting prompt is used. "MI" means it is the multi-image version dataset from DEMON Li et al. (2024e).

| ID | Type | Postion | Prompt |
|----|------|---------|--------|
| 1 | VQA | Tail | Answer the question with a single word (or phrase). |
| 2 | VQA | Head | Hint: Please answer the question and provide the final answer at the end. |
| 3 | VQA (Yes/No) | Tail | Answer the question with Yes or No./Yes or No?/... |
| 4 | Choice | Tail | Answer with the given letter directly |
| 5 | Choice (Option Letter) | Tail | Answer with the option letter from the given choices directly. / Please respond with only the letter of the correct answer. |
| 6 | Choice (Option Letter) | Head | Hint: Please answer the question and provide the correct option letter, e.g., A, B, C, D, at the end. |
| 7 | Region Caption | All | Provide a short description for this region. |
| 8 | Grounding | All | Provide the bounding box coordinate of the region this sentence describes. |
| 9 | Breif Caption | All | Provide a one-sentence caption for the provided image./Create a compact narrative representing the image presented./... |
| 10 | Screen Summarization | All | Summarize the main components in this picture./Provide a detailed account of this screenshot./... |
| 11 | Detailed Caption | All | Describe this image in detail./Explain the visual content of the image in great detail./... |
| 12 | Science Books | All | Here is a diagram figure extracted from some Grade 1 - 6 science books.\nPlease first describe the content of this figure in detail, including how the knowledge visually displayed in the diagram.\nThen start with a section title \"related knowledge:\", briefly and concisely highlight the related domain knowledge and theories that underly this diagram. Note that you do not need to provide much detail. Simply cover the most important concepts. |
| 13 | Information Extraction | Head | Provide the requested information directly. |
| 14 | Graph Sumarization | All | Please clarify the meaning conveyed by this graph./Explain what this graph is communicating./... |
| 15 | Photo Sumarization | All | Highlight a few significant elements in this photo./Mention a couple of crucial points in this snapshot./... |
| 16 | Chart Sumarization | All | What insights can be drawn from this chart?/Explain the trends shown in this chart./... |
| 17 | OCR | Head | OCR this image section by section, from top to bottom, and left to right. Do not insert line breaks in the output text. If a word is split due to a line break in the image, use a space instead |
| 18 | Diagram Linkage | All | Dissect the diagram, highlighting the interaction between elements./Interpret the system depicted in the diagram, detailing component functions./... |
| 19 | Code Generation | All | Compose the HTML code to achieve the same design as this screenshot. |
| 20 | Choice (with Reasoning) | Head | First perform reasoning, then finally select the question from the choices in the following format: Answer: xxx. |
| 21 | Math Computing | Tail | Round computations to 2 decimal places. |
| 22 | LaTeX OCR | All | Please write out the expression of the formula in the image using LaTeX format. |
| 23 | Text Reading | All | What is written in the image? Answer this question using the text in the image directly./Read and list the text in this image. |
| 24 | Choice (Full Option) | Tail | Please provide your answer by stating the letter followed by the full option. |

Table 18: The information of formatting prompts for Single-Image data. The "Position" means the position of the formatting prompt in the prompt where "All" means the formatting prompt is the prompt. Sometimes, there are multiple prompts of the same meaning. In this case, the prompt column is fomatted as "Prompt1/Prompt2/...".

| ID | Type | Postion | Prompt |
|---|---|---|---|
| | | | Video |
| 1 | Choice (Option Letter) | Tail | Answer with the option letter from the given choices directly. / Please respond with only the letter of the correct answer. |
| 2 | Choice (Full Option) | Tail | Please provide your answer by stating the letter followed by the full option. |
| | | | Multi-Image |
| 3 | Open-Ended | Head | What's the difference between 2 images? |
| 4 | Open-Ended | Head | Given the stories paired with the first several images, can you finish the story based on the last image?/With the narratives paired with the initial images, how would you conclude the story using the last picture?/... |
| 5 | Multi-Choice | Head | Here is a Raven's Progressive Matrice in a three-by-three form. You are provided with the first eight elements in eight images, please select the last one from four choices following the structural and analogical relations. |
| 6 | Multi-Choice | All | There are ten possible explanations for the ten different answers to a VQA: ... I will give you two sets of pictures, questions, and answers to determine if they belong to the same 'Question-Answer Differences'. You must choose your answer from the Choice List. |
| 7 | Open-Ended | Head | This is a 3D scenario. |
| 8 | Open-Ended | Head | I will provide you several images and a question, your job is to seek information in the slide and answer the question correctly./Based on the images, please answer the following question./... |
| 9 | Multi-Choice | Head | Provided with a series of diagrams from a textbook, your responsibility is to correctly answer the following question. You must choose your answer from the Choice List./Using a selection of textbook diagrams, your task is to provide an accurate response to the subsequent query. You must choose your answer from the Choice List./... |
| 10 | Open-Ended | Head | Given six images taken from different cameras on a street view car, your task is to answer questions about the depicted scene. You must choose your answer from the Choice List. /Upon receiving six photographs captured from various cameras on a street-view car, your responsibility is to provide accurate responses to questions about the scene. You must choose your answer from the Choice List. /... |
| 11 | Multi-Choice | Head | I will provide you with two sets of pictures, each of which shows an object in the opposite state. Can you tell me if the states of these two sets of pictures are the same? You must choose your answer from the Choice List. /I have two sets of pictures that show an object in opposite states. Can you tell me if the states of these two sets of pictures are the same? You must choose your answer from the Choice List. /... |
| 12 | Multi-Choice | Head | Are the following four images of the same class? You must choose your answer from the Choice List. /Do the following four images belong to the same category? You must choose your answer from the Choice List. /... |
| 13 | Multi-Choice | Head | Are these two workpieces the same type?/Are these two workpieces of the same kind?/... |
| 14 | Multi-Choice | Head | Presented with a textual recipe tutorial, your task is to scrutinize it carefully and select the image that is incoherent in the provided sequence of images. You must choose your answer from the Choice List. /Given a text-based recipe guide, your responsibility is to meticulously review it and identify the image that doesn't fit in the following sequence of images. You must choose your answer from the Choice List. /... |
| 15 | Multi-Choice | Head | I will give you a series of comic panels. The dialogue box of the last panel is masked. Can you choose the most relevant one from the candidates? You must choose your answer from the Choice List. /Given previous full panels and one masked panel, your job is to select the most appropriate dialogue among four candidates. You must choose your answer from the Choice List. /... |
| 16 | Open-Ended | Head | Give you a main goal, your job is to figure out what to do now by looking at current envirments. Your past views as well as decisions are also provided./Given a primary objective and your current surroundings, use your previous decisions and perspectives to determine your next move./... |
| 17 | Multi-Choice | Head | I will give you two pictures of the book cover. Please look at the pictures and answer a question You must choose your answer from the Choice List. /I will provide you with two images of the book cover. Please examine the images and answer a question. You must choose your answer from the Choice List. /... |
| 18 | Multi-Choice | Head | I will give you some pictures, and each group of pictures will correspond to a question. Please answer it briefly. You must choose your answer from the Choice List. /For each group of pictures, there is a question. Please give a short answer to it. You must choose your answer from the Choice List. /... |
| 19 | Open-Ended | Head | Please give a editing Request to describe the transformation from the source image to the target image./What is the correct image edit instruction that can transfrom the source image to target image?/... |
| 20 | Open-Ended | Head | What's the difference between 2 images? /Identify the alterations between these two images. /... |
| 21 | Open-Ended | Head | What's the difference between 2 birds? /Identify the alterations between these two birds. /... |
| 22 | Open-Ended | Head | What's the difference between 2 images? /Identify the alterations between these two images. /... |
| 23 | Open-Ended | Head | Given the stories paired with the first several images, can you finish the story based on the last image?/With the narratives paired with the initial images, how would you conclude the story using the last picture?/... |
| 24 | Open-Ended | Head | Given the stories paired with the first several images, can you finish the story based on the last image?/With the narratives paired with the initial images, how would you conclude the story using the last picture?/... |
| 25 | Open-Ended | Head | Given the stories paired with the first several images, can you finish the story based on the last image?/With the narratives paired with the initial images, how would you conclude the story using the last picture?/... |
| 26 | Multi-Choice | All | Answer the following multiple-choice question: Here is a statement describing 2 images: ... Is it true or false? |

Table 19: The information of formatting prompts for One-Vision data. The "Position" means the position of the formatting prompt in the prompt where "All" means the formatting prompt is the prompt. Sometimes, there are multiple prompts of the same meaning. In this case, the prompt column is fomatted as "Prompt1/Prompt2/...".

### C.4 Policy Information and Reproducibility

We will open-source most of the public datasets we used. These images and data are already publicly available for academic research; we incorporated them and converted the format for our use. However, a small portion of our data sources related to user data and those obtained using the Azure OpenAI Service cannot be directly released due to company policy. We will provide the exact data YAML files used in the final reproduction scripts and will offer reproducible experimental scripts, training logs, and final version checkpoints using fully public data as our compute resources allow.

