# OpenReview forum: "LLaVA-OneVision: Easy Visual Task Transfer"
_TMLR — Accepted by TMLR_

### Review · Reviewer_YF8X · 2024-11-02

**Summary Of Contributions:**

This paper introduces a strong family of large multimodal models (LMMs) up to 72B parameters, outperforming most current baselines. The core contribution is in training a single model that excels across single-image, multi-image, and video tasks. The paper provides details on training processes, data curation, and extensive evaluation results.

**Audience:**

Yes

**Broader Impact Concerns:**

The model is a text generation model and based on LLMs, it can generate harmful or toxic text if there is no explicit mechanism to prevent it from doing so.

**Claims And Evidence:**

No

**Requested Changes:**

Please check the weaknesses section. The work missing ablation study to understand how the model can attain such performance.

**Strengths And Weaknesses:**

**Strengths**:

* The model demonstrates strong improvements over existing models and across various scales.
* The paper is well-structured and clearly presents the stages involved in building such a model.
* Extensive evaluation across many benchmarks provides a comprehensive view of the model's capabilities.

**Weaknesses**:

1. My main concerns about this paper is that it focus only on showing the final model scores, without providing sufficient ablation studies or analyses on main parts of the training pipeline, for example:

- How does the data mixture used here compare to other datasets, such as Cambrain or Cauldron? Is there a strategic assembly of the training mixture, and how does each dataset contribute to the final model’s performance?
- The model is trained in four stages. Are all stages essential, or could the first or second stages be omitted without significant loss in performance?
- Is training the vision encoder necessary, and if so, to what extent does it impact the results?
- The authors selected Qwen as the LLM —are the results sensitive to this choice? It would be valuable to understand the impact of different LLMs on final scores.
- Each stage is trained for only one epoch—is this due to performance saturation after the first epoch?
- Performance comparisons between the Anyres and Higher Anyres models are not addressed.

2. A primary contribution is the model's task transfer abilities. However, this has been a topic in several prior works, which are not cited here. For instance, [1] examines task transfer from image to video, incorporating curriculum learning similar to the approach here. Citing and discussing relevant literature is important to properly position the paper within existing research.

[1] Shukor, Mustafa, et al. "UnIVAL: Unified Model for Image, Video, Audio and Language Tasks." TMLR 2023.

---

> ### Author Response · Authors · 2024-12-19
> **Response to Reviewer YF8X - Part 1**
>
> Thank you for your thoughtful and constructive feedback. We appreciate the time and effort you put into reviewing our work. Below, we address the concerns and suggestions raised:
>
> ---
>
> **Comment:** Ablation Studies and Analyses
>
> **Response:** We agree that detailed ablation studies can provide deeper insights into the model's performance and training dynamics. To address your concerns:
>
> 1. **Data Mixture Comparison:**
>
>     While we acknowledge the importance of comparing our dataset mixture with Cauldron and Cambrian to demonstrate its effectiveness, making direct comparisons is challenging due to computational constraints. Since we incorporated their datasets during our collection phase, and as mentioned in our response to Reviewer xVoN, conducting the single-image and OneVision stage training requires 32 H100 GPUs for several days.
>
> 2. **Training Stages:**
>
>     Since Stage 1's pretraining projector has become an essential component widely adopted since LLaVA-1.0, we typically don't perform ablations on this part. The results for single-image and OneVision stages are already presented in Table 3 of the main paper.
>
>     To complement this, we provide a minimal ablation study on Stage 1.5, which uses large amounts of high-quality recaptioned data to improve performance. Below, we refer the table from our response to Reviewer UqMZ.
>
>     *Table 1. Enhanced Performance with Recaptioned Data*
>
>     | **Training Data** |  | **Avg.** | **AI2D** | **ChartQA** | **DocVQA** | **InfoVQA** | **MathVista** | **MME** | **LLaVA-W** | **ScienceQA** | **Image-DC** |
>     | --- | --- | --- | --- | --- | --- | --- | --- | --- | --- | --- | --- |
>     | **Stage 1.5** | **Stage 2** | **-** | **test** | **test** | **val** | **val** | **testmini** | **-** | **-** | **IMG** | **EN** |
>     | - | 790K | 68.2 | 66.9 | 65.2 | 75.3 | 36.7 | 35.6 | 65.1 | 79.8 | 69.7 | 88.2 |
>     | 118K (ReCap) | 790K | 68.6 | 66.9 | 66.6 | 75.5 | 36.6 | 36.1 | 65.7 | 79.7 | 71.0 | 87.6 |
>     | 558K (ReCap) | 790K | 69.4 | 70.1 | 67.8 | 76.9 | 39.4 | 36.2 | 65.1 | 79.4 | 71.5 | 88.2 |
>     | 3M (Recap) | 790K | 70.7 | 72.7 | 68.3 | 77.7 | 38.1 | 38.6 | 65.7 | 80.1 | 72.0 | 90.4 |
> 3. **Vision Encoder Training:** We tested the impact of training the vision encoder versus freezing it during training. The results show that fine-tuning the vision encoder improves performance, particularly for high-resolution tasks. This improvement makes sense because our Anyres strategy crops original large images into small tiles, creating inputs that differ significantly from the whole-image data the vision encoder was originally trained on. Consequently, training the vision encoder leads to better performance on tasks requiring high-resolution processing.
>
>     *Table 2. Freeze versus Unfreeze Vision Encoder*
>
>     | **Training Data** | **Vision Encoder** | **LLM Decoder** | **Avg.** | **AI2D** | **ChartQA** | **DocVQA** | **InfoVQA** | **MathVista** | **LLaVA-W** | **ScienceQA** |
>     | --- | --- | --- | --- | --- | --- | --- | --- | --- | --- | --- |
>     | 790K | SO400M (freeze) | Qwen2-7B-Instruct | 57.8 | 61.7 | 58.5 | 70.4 | 33.2 | 33.6 | 77.8 | 69.7 |
>     | 790K | SO400M (unfreeze) | Qwen2-7B-Instruct | 61.3 | 66.9 | 65.2 | 75.3 | 36.7 | 35.6 | 79.8 | 69.7 |

---

> > ### Author Response · Authors · 2024-12-19
> > **Response to Reviewer YF8X - Part 2**
> >
> > 4. **LLM Choice Sensitivity:** We agree that exploring the impact of different LLMs is essential. Our ablation experiments comparing alternative LLMs (Qwen-1.5, Qwen-2, and LLaMA-3) show that the Qwen-2 model achieves better overall performance, though improvements vary across tasks. We also conducted scaling experiments with Qwen-1.5 models ranging from 0.5B to 110B parameters, as Qwen-1.5 offers the most comprehensive range of model sizes compared to other series that only offer 7B or 70B variants.
> >
> >     *Table 3. Comparison of Different LLM*
> >
> >     | **Training Data** | **LLM Decoder** | **Avg.** | **AI2D** | **ChartQA** | **DocVQA** | **InfoVQA** | **MathVista** | **LLaVA-W** | **ScienceQA** |
> >     | --- | --- | --- | --- | --- | --- | --- | --- | --- | --- |
> >     | 790K | Qwen1.5-7B-Instruct | 59.1 | 64.3 | 64.7 | 73.2 | 35.9 | 32.1 | 74.9 | 68.9 |
> >     | 790K | LLaMA-3-8B | 60.9 | 66.7 | 64.8 | 76.2 | 34.3 | 33.7 | 80.5 | 70.1 |
> >     | 790K | Qwen2-7B-Instruct | 61.3 | 66.9 | 65.2 | 75.3 | 36.7 | 35.6 | 79.8 | 69.7 |
> >
> >     *Table 4. Comparison of Different Sizes LLM*
> >
> >     | **LLM Decoder** | **Avg.** | **AI2D** | **ChartQA** | **DocVQA** | **MathVista** | **MMMU** | **LLaVA-W** | **ScienceQA** |
> >     | --- | --- | --- | --- | --- | --- | --- | --- | --- |
> >     | Qwen-1.5 | **-** | **test** | **test** | **val** | **testmini** | **dev** | **-** | **IMG** |
> >     | 0.5B | 52.8 | 49.4 | 54.8 | 63.4 | 28.1 | 29.4 | 61.7 | 60 |
> >     | 1.8B | 57.6 | 59.5 | 58.2 | 67.6 | 29.3 | 32.8 | 69.7 | 66 |
> >     | 4B | 63.7 | 68.6 | 65.2 | 73.8 | 34.5 | 36.4 | 76.1 | 70.8 |
> >     | 7B | 65.2 | 73.5 | 68.5 | 75.7 | 32.1 | 37.4 | 76.4 | 72.5 |
> >     | 14B | 70.7 | 75.8 | 71.5 | 80.8 | 41.2 | 43.3 | 86.6 | 77.5 |
> >     | 32B | 72.7 | 76.3 | 74 | 79.8 | 42.6 | 48.9 | 90.8 | 81.5 |
> >     | 72B | 74 | 77.4 | 77 | 84.4 | 46.6 | 46.4 | 89.2 | 83.9 |
> >     | 110B | 76 | 80.4 | 79.7 | 85.7 | 49 | 49.1 | 90.4 | 83.2 |
> > 5. **One Epoch Training:** The decision to train each stage for one epoch was based on performance saturation according to our experience and observation during preliminary experiments. During training, we observed that when the model entered the second epoch, the SFT loss showed a steep decline, suggesting that the model began to optimize through direct memorization rather than knowledge learning. Training for additional epochs resulted in marginal improvements at significantly increased computational costs. These findings will be clarified and expanded in the revised version.
> > 6. **Performance Comparison of Anyres and Higher Anyres:** Thanks for addressing it. We did provide a more detailed analysis in together of the response to Reviewer UqMZ in Table 1, including specific task-wise comparisons to better highlight the trade-offs and advantages of each approach.
> >
> > ---
> >
> > **Comment:** Positioning Within Prior Work
> >
> > **Response:** We appreciate the reference to UnIVAL[1] and acknowledge the need to position our work more clearly within existing research. We will include relevant citations and discuss how our task transfer approach differs from and builds upon prior work.
> >
> > [1] Shukor, Mustafa, et al. "Unified model for image, video, audio and language tasks." TMLR 2023.
> >
> > ---
> >
> > **Comment:** Broader Impact
> >
> > **Response:** We recognize the potential risks of harmful or toxic text generation. While this concern is broadly applicable to models based on LLMs, we believe the safety risks are mitigated for two main reasons:
> >
> > (1) Open-source Data: Our training data is sourced from the open-source community and has been scrutinized by community users and releasing organizations.
> >
> > (2) Open-source Model: Our models are open-source with publicly available weights, enabling safety researchers to inspect, test, and report potential safety concerns.
> >
> > We consider safety-related issues to be critically important and will emphasize this aspect in our discussion and revised manuscript.
> >
> > ---
> >
> > We hope that these updates will address your concerns and improve the clarity and depth of the paper. Thank you again for your valuable feedback. Let me know if further adjustments or additional details are needed!

---

> > > ### Comment · Reviewer_YF8X · 2024-12-22
> > >
> > > Thanks for the additional experiments and clarifications. The authors addressed most of my concerns. I highly recommend adding these experiments to the final paper.

---

### Review · Reviewer_UqMZ · 2024-11-18

**Summary Of Contributions:**

Large multimodal models (LMMs) have advanced multimodal learning research. The recent work, LLaVA-Next, builds prototypes of LMMs and brings insights about data, models, and visual representations for LMMs. Inspired by these insights, this work introduces LLaVA-OneVision (LLaVA-OV) which can handle more variations of visual inputs (e.g., table/chart/multiple videos, etc.). LLaVA-OV processes different visual inputs (images/multi-images/videos) via the proposed Higher AnyRes strategy. The model is pretrained on a mixture of existing datasets and newly generated datasets with a curriculum learning approach. LLaVA-OV is evaluated on LMMs-Eval. The experimental results show that LLaVA-OV-72B can outperform the existing open-sourced LMMs and reach the performance between GPT-4V and GPT-4o on majority of the benchmarks. At last, the work discusses that LLaVA-OV emerges several task transfer capabilities on more versatile visual inputs.

**Audience:**

Yes

**Broader Impact Concerns:**

There are no concerns on the ethical implications.

**Claims And Evidence:**

Yes

**Requested Changes:**

1. [Minor] There is only one line of content on Page 4. It will be better if this page only shows the two existing figures.
2. [Minor] Missing citations for VQAv2, GQA, VG on Page 6 (in Section 4.2).
3. [Minor] What is “LCS” in Table 1?
4. [Minor] There are many digits to read into on the main result tables. Is it possible to summarize some of the results with average scores?
5. [Typo] “responces” -> “responses” on Page 8
6. [Typo] “the the” -> “the” on Page 9

**Strengths And Weaknesses:**

**Strengths**
1. The work and newly collected data will be open-sourced which are big contributions to the community.
2. The experimental results show that LLaVA-OV perform competitively against existing open-sourced LMMs and even GPT-4V/-4o. Its performance on video benchmarks is particularly strong.
3. The transfer task capabilities show great potentials of LLaVA-OV and also provide inspirations for future research.
4. The paper is clearly written and well-organized.


**Weaknesses**
1. There are lack of ablation studies to verify the contributions of the newly proposed components, such as Higher AnyRes strategy, newly generated data, more use of data, and the curriculum learning during pretrainnig. However, it is also understandable that such ablation studies are too resource-wise intensive.
2. On Table 3, though LLaVA-OV-72B can outperform all the open-sourced models, but the comparisons are not fair because of the large size of LLM (72B). Actually, InternVL-8b can outperform LLaVA-OV-7B on lots of single-image benchmarks. Can any patterns be found from there? One speculation is that Higher AnyRes strategy leads to long sequences for single image inputs (i.e., a video is 6272 tokens, but a single image is 7290 tokens), thus may lead to lower density of information for each token. Thus, LLaVA-OV needs to aggregate information from a long sequence for single-image problems, which makes the problems possibly harder.
3. Though LLaVA-OV has improved visual processing ability, how about the language processing/understanding after the training with multimodal data? This is a meaningful and important question for LMMs.
4. The task transfer capabilities are generally discussed in Section 7. It will be better if a small set of data points can be collected in order to evaluate LLaVA-OV with other LMMs to further demonstrate the emergent capabilities of LLaVA-OV.

---

> ### Author Response · Authors · 2024-12-19
> **Response to Reviewer UqMZ - Part 1**
>
> We sincerely thank the reviewer for their thoughtful comments and constructive feedback. Below, we address each point raised.
>
> ---
>
> **Comment:** More Ablation Studies
>
> **Response:** We appreciate the suggestion regarding ablation studies. While resource constraints limit exhaustive evaluations, we include focused ablation analyses in to provide better insights on our design.
>
> We first provide the analysis on the influence of resolution and token count on training time, with results summarized below.
>
> *Table 1. Impact on Max. #Grids in AnyRes and Max. #Tokens.*
>
> | **Max. #Grids** | **Max. #Tokens** | **Training Time** | **AI2D** | **ChartQA** | **DocVQA** | **InfoVQA** | **OK-VQA** | **POPE** | **MMMU** |
> | --- | --- | --- | --- | --- | --- | --- | --- | --- | --- |
> |  |  |  | test | test | val | val | val | Test/F1-score | val |
> | 2x2 | (4+1)*729 | 6H30M | 51.1 | 49.2 | 58.8 | 25.7 | 36.5 | 85.4 | 28.2 |
> | 4x4 | (4+1)*729 | 7H30M | 52.8 | 49.4 | 58.1 | 26.0 | 36.0 | 85.8 | 28.6 |
> | 5x5 | (4+1)*729 | 7H50M | 52.4 | 49.6 | 57.6 | 26.9 | 36.5 | 86.1 | 28.4 |
> | 6x6 | (4+1)*729 | 8H05M | 52.7 | 50.1 | 56.7 | 27.1 | 35.9 | 85.9 | 28.3 |
> | 6x6 | (9+1)*729 | 11H14M | 52.7 | 55.8 | 62.7 | 26.7 | 42.0 | 86.1 | 29.3 |
> | 6x6 | (16+1)*729 | 13H10M | 52.7 | 56.1 | 62.2 | 27.1 | 42.5 | 87.4 | 27.4 |
>
> Our analysis reveals key insights about the AnyRes token strategy:
>
> - Increasing the maximum AnyRes grid size from 2×2 to 6×6 for higher resolution yielded better performance on tasks that require detailed image analysis, particularly InfoVQA.
> - While higher resolution only modestly impacts training time, increasing token count creates substantial computational overhead. Yet maintaining a 6×6 grid while increasing tokens notably improves OCR capabilities, as shown by better performance on ChartQA and DocVQA. These results indicate that prioritizing resolution over token count provides the optimal balance for enhancing visual representations.
>
> We then study the impact on adding high-quality detailed caption data as in pretraining stage.
>
> *Table 2. Enhanced Performance with Recaptioned Data*
>
> | **Training Data** |  | **Avg.** | **AI2D** | **ChartQA** | **DocVQA** | **InfoVQA** | **MathVista** | **MME** | **LLaVA-W** | **ScienceQA** | **Image-DC** |
> | --- | --- | --- | --- | --- | --- | --- | --- | --- | --- | --- | --- |
> | **Stage 1.5** | **Stage 2** | **-** | **test** | **test** | **val** | **val** | **testmini** | **-** | **-** | **IMG** | **EN** |
> | - | 790K | 68.2 | 66.9 | 65.2 | 75.3 | 36.7 | 35.6 | 65.1 | 79.8 | 69.7 | 88.2 |
> | 118K (ReCap) | 790K | 68.6 | 66.9 | 66.6 | 75.5 | 36.6 | 36.1 | 65.7 | 79.7 | 71.0 | 87.6 |
> | 558K (ReCap) | 790K | 69.4 | 70.1 | 67.8 | 76.9 | 39.4 | 36.2 | 65.1 | 79.4 | 71.5 | 88.2 |
> | 3M (Recap) | 790K | 70.7 | 72.7 | 68.3 | 77.7 | 38.1 | 38.6 | 65.7 | 80.1 | 72.0 | 90.4 |
>
> We observe several key insights:
>
> 1. Models trained with recaptioned data (ReCap) datasets demonstrate enhanced performance on tasks that require detailed image descriptions and document understanding.
> 2. The regenerated captions (ranging from 118K to 3M) show better scaling behavior than the original captions and consistently improve model performance across various metrics.
> 3. With ReCap data, full-model training proves more effective than projector tuning since larger model capacity is needed to process high-quality knowledge. This approach yields significant improvements in metrics including AI2D, DocVQA, ChartQA, InfoVQA, and ScienceQA.
>
> The above two sections provide our insights on the AnyRes-Max strategy and high-quality mid-stage pretraining data. Due to computational constraints, we cannot ablate additional aspects. The remaining design elements are primarily adapted from open-source community works such as Qwen-VL[1], InternVL, Cambrian-1, and X-Composer.
>
> [1] Bai, Jinze, et al. "Qwen-vl: A frontier large vision-language model with versatile abilities." *arXiv preprint arXiv:2308.12966* (2023).
>
> [2] Chen, Zhe, et al. "Internvl: Scaling up vision foundation models and aligning for generic visual-linguistic tasks." *Proceedings of the IEEE/CVF Conference on Computer Vision and Pattern Recognition*. 2024.
>
> [3] Tong, Shengbang, et al. "Cambrian-1: A fully open, vision-centric exploration of multimodal llms." *arXiv preprint arXiv:2406.16860* (2024).
>
> [4] Dong, Xiaoyi, et al. "Internlm-xcomposer2-4khd: A pioneering large vision-language model handling resolutions from 336 pixels to 4k hd." *arXiv preprint arXiv:2404.06512* (2024).

---

> ### Author Response · Authors · 2024-12-19
> **Response to Reviewer UqMZ - Part 2**
>
> **Comments:** Comparison with InternVL-2-8b
>
> **Response:**  We acknowledge that many current SOTA models have demonstrated their effectiveness across benchmarks, and improving upon these benchmarks is challenging.
>
> Here we should note that:
>
> 1. Regarding contributions to the open-source community: We have fully open-sourced all our training data and training processes. This data is available to help the open-source community build better LMM models, while the Intern-2 series has not achieved complete data open-sourcing. Additionally, our training process (as detailed in our response to Reviewer xVoN) demonstrates that our model is highly economical to train, requiring only hundreds of GPUs and a few days to build a high-performing multimodal model.
> 2. On benchmark diversity: While InternVL-2 and X-Composer models show strong performance on traditional benchmarks with fixed response formats (like AI2D, ChartQA, DocVQA), our model demonstrates clear advantages on benchmarks requiring more flexible responses and real-world user queries (such as Vibe-Eval, LLaVA-W, LLaVA-Wilder). Similarly, for mathematical reasoning benchmarks like MathVista, our model achieves performance closer to GPT4's level.
>
> We greatly value the reviewer's point about our token strategy potentially affecting single-image performance due to increased token count. We agree this is a valuable research direction worthy of further exploration.
>
> Furthermore, based on the InternVL series reports, their image input tokens actually exceed those of LLaVA-OneVision due to their use of more tiles for image representation. Therefore, these conclusions remain uncertain, and final performance may be more influenced by the volume of OCR, Chart, and Document training data.
>
> ---
>
> **Comments:** Language Processing Ability
>
> **Response:** The concern about potential degradation of language understanding capabilities after multimodal training is valid. We provide evaluation results on LLaVA-OV on language benchmarks to demonstrate that the model maintains its language understanding capabilities. Some of the results are from [1]
>
> | **Model** | **Backbone LLM** | **MMLU** | **GSM8K** | **MATH** | **HumanEval** | **Avg. Accuracy** |
> | --- | --- | --- | --- | --- | --- | --- |
> | Qwen2-72B-Instruct | N/A | 82.3 | 91.1 | 59.7 | 86.0 | 79.8 |
> | Llama-3-70B-Instruct | N/A | 82.0 | 93.0 | 51.0 | 81.7 | 76.6 |
> | LLaVa-OneVision 72B | Qwen2-72B-Instruct | 80.6 | 89.9 | 49.2 | 74.4 | 73.5 |
> | InternVL-2-Llama3-76B  | Llama-3-70B-Instruct | 78.5 | 87.1 | 42.5 | 71.3 | 69.9 |
>
> The results demonstrate that this performance trade-off occurs in both LLaVA-OneVision and other powerful multimodal models like InternVL-2. This appears to be an unavoidable challenge in current multimodal work. For example, LLaMA-3.1-Vision [2] addresses this by using very large adapter parameters without training the LLM to maintain its performance. However, this approach increases parameter consumption (for instance, LLaMA-3.1-Vision-90B, derived from LLaMA-3.1-70B, requires nearly 20B additional parameters for encoders and adapters).
>
> [1] Dai, Wenliang, et al. "Nvlm: Open frontier-class multimodal llms." *arXiv preprint arXiv:2409.11402* (2024).
>
> [2] https://ai.meta.com/blog/meta-llama-3-1/
>
> ---
>
> **Comment:** Task Transfer Evaluation
>
> **Response:** We acknowledge the need for empirical evidence to support our observations on task transfer capabilities.
>
> We agree that task transfer is a core capability for evaluating LMMs, we refer the audience to check the Response to Reviewer xVoN.
>
> ---
>
> We greatly appreciate the reviewer’s constructive feedback and have taken all suggestions into account to improve the manuscript further. Thank you again for your time and effort.

---

> > ### Comment · Reviewer_UqMZ · 2024-12-21
> >
> > Thank you for the response, and it addresses majority of my concerns. These new results can add extra values to the paper, so I highly recommend the authors to include and discuss them in the final version, especially the ablation studies and the language processing ability ones. I will keep my rating.

---

### Review · Reviewer_xVoN · 2024-11-21

**Summary Of Contributions:**

1. This paper presents LLaVA-OneVision, a family of large multimodal models that aims to achieve strong performance across single-image, multi-image, and video understanding tasks through a unified architecture. The key contributions include:
2. The design of higher AnyRes with Bilinear Interpolation is a good technique that enables more vision information being processed by the LLM. This streamlined architecture design enables effective transfer learning between modalities, particularly leveraging strong image understanding to benefit video tasks.
3. A comprehensive curation of 3.2M single-image and 1.6M multi-modal training samples, with careful consideration of data quality and task distribution.
4. Detailed transfer learning analysis of LLaVA-OneVision in all kinds of settings.

**Audience:**

Yes

**Claims And Evidence:**

Yes

**Requested Changes:**

1. Please provide more details of the curated training datasets as stated in the weaknesses.
2. Please provide the training details like GPU hours, and other information, if allowed.
3. In the second paragraph of the related works, "For the latter, our are motivated by FLAN ..." seems to a grammar error.
4. It seems number of tokens per frame is 196, which is lower than the number of tokens per image in the multi-image scenarios. Are there any explanations of this design and the experiments supports the explanations?

**Strengths And Weaknesses:**

**Strengths**
1. A family of LLaVA-OneVision models scaling from 7B to 72B size.
2. A comprehensive curation of 3.2M single-image and 1.6M multi-modal training samples, with careful consideration of data quality and task distribution
3. Comprehensive experiments on all the 3 major vision scenarios, single-image, multi-image, and video understanding. And LLaVA-OneVision shows strong results
4. Insights about the transfer learning in different settings.

**Weaknesses**
1. The paper lacks statistical details of the curated datasets. For example, the token length distributions for single-image datasets, the number of multiple images in the multi-image datasets, and the video statistics time ranges, etc. The existing detailed information in Table 16 only contains the number of samples and their prompt template ID but missing these important information to help people better understand the datasets's advantages and disadvantages
2. Lack of training details of LLaVA-OneVision models, such as GPU hours, and number of GPUs, etc.
3. While section 7 indeed talks about some transfer ability of LLaVA-OneVision in different scenarios, some of them might lack comprehensive evaluation to benchmarking the abilities. For example, Video-to-Video Difference, set-of-mark prompting and visual prompting might lack specific benchmark for the ability.

---

> ### Author Response · Authors · 2024-12-19
> **Response to Reviewer xVoN - Part 1**
>
> We appreciate the detailed feedback and the insightful suggestions for improving the paper. Below, we address each of the reviewer's comments:
>
> ---
>
> ### **Statistical Details of Curated Datasets**
>
> **Comment:** The paper lacks statistical details of the curated datasets, such as token length distributions, number of multiple images, and video statistics like time ranges.
>
> **Response:**
>
> We appreciate the suggestion to include more comprehensive statistical information about the datasets. To address this:
>
> 1.	**Token Length Distributions for Single-Image Datasets:**
>
> We will include histograms of token length distributions for single-image datasets in the revised version of the paper. These distributions will highlight the variability in the dataset and its potential impact on model performance. Here we provide the information about the tokens (including image tokens processed in `anyres_max_9` strategy) in pure text.
>
> - **Max Tokens:** 22768
> - **Min Tokens:** 948
> - **Mean Tokens:** 3038.5
>
> 2.	**Multi-Image Dataset Details:**
>
> For the multi-image datasets, we will add the statistics on the number of images per sample. This information will clarify how multi-image contexts are constructed and provide insights into the dataset’s structure.
>
> - **Max Images:** 12
> - **Min Images:** 2
> - **Mean Images:** 4.7
>
> 3.	**Video Dataset Duration Statistics:**
>
> We have calculated the video duration statistics as follows:
>
> - **Max Duration:** 29.97 seconds
> - **Min Duration:** 5.00 seconds
> - **Mean Duration:** 18.14 seconds
>
> However, these statistics exclude the `sharegpt4video_255000` dataset for the following reasons:
>
> - This dataset is unique in that it samples exactly 10 frames regardless of the video’s length, making duration statistics less meaningful.
> - Additionally, “sharegpt4video_255000” is not part of the datasets included in LLaVA-Video but rather an academic dataset we used for comparison.
>
> 4.	**Frame Sampling Strategy:**
>
> We also clarify our frame extraction process for the video datasets:
>
> - Frames are extracted at 1 frame per second (fps) to create the initial set of frames (frames_1).
> - If the total frames exceed 32, we uniformly sample 32 frames from frames_1 to ensure consistency.
> - If the total frames are fewer than 32, the video duration typically corresponds to a few seconds with proportionally fewer frames.
>
> These additional details will be incorporated into the revised manuscript, specifically in Table 16 and the associated discussions. We believe these updates will help readers better understand the datasets’ strengths, limitations, and their impact on model performance. Thank you for bringing this to our attention.
>
> ---
>
> ### **Training Details of LLaVA-OneVision Models**
>
> **Comment:** Lack of training details such as GPU hours, number of GPUs, etc.
>
> **Response:** We acknowledge the importance of transparency in training details. We now provide comprehensive information about our computational resources and training process:
>
> - Our computing infrastructure consists of  ~380 A100-HBM and 128 H100-HBM GPUs.
> - Training requirements vary by model size:
>     - 0.5B Model (Total: 55-64 hours)
>         - Stage 1: 32 A100 GPUs for 1 hour
>         - Stage 1.5: 32 A100 GPUs for 15-20 hours
>         - Stage 2 Single Image: 32 H100 GPUs for 20-24 hours
>         - Stage 2 OneVision: 32 H100 GPUs for 20 hours
>     - 7B Model (Total: 58-66 hours)
>         - Stage 1: 32 A100 GPUs for 1 hour
>         - Stage 1.5: 256 A100 GPUs for 18 hours
>         - Stage 2 Single Image: 256 A100 GPUs for 20-24 hours
>         - Stage 2 OneVision: 128 H100 GPUs for 20-24 hours
>     - 72B Model (Total: 180 hours)
>         - Stage 1: 32 A100 GPUs for 1 hour
>         - Stage 1.5: 256 A100 GPUs for 60 hours
>         - Stage 2 Single Image: 256 A100 GPUs for 80 hours
>         - Stage 2 OneVision: 128 H100 GPUs for 40 hours
>
> The reported training times include system maintenance, GPU failures, and restart periods. In summary, the maximum training durations for the 0.5B/7B/72B models are 64/66/180 hours respectively, requiring either 32/128/128 H100 GPUs or 32/256/256 A100 GPUs.
>
> We believe these computational requirements are notably lower than those of comparable multimodal models achieving similar performance.
>
> These details provide readers with a clear understanding of the computational costs and scalability involved in training the LLaVA-OneVision models.

---

> > ### Author Response · Authors · 2024-12-19
> > **Response to Reviewer xVoN - Part 2**
> >
> > ### **Comprehensive Benchmarking of Transfer Abilities**
> >
> > **Comment:** Some transfer ability evaluations lack comprehensive benchmarks (e.g., Video-to-Video Difference, set-of-mark prompting, visual prompting).
> >
> > **Response:** We appreciate this observation. To address this, we have conducted additional evaluations with specific benchmarks for these capabilities:
> >
> >
> > *Table 1. Visual Prompting Task*
> >
> > | **Model** | **Visual7W** | **PointQA-LookTwice** |
> > | --- | --- | --- |
> > | ViP-LLaVA-7B* | 86.6 | 71.3 |
> > | LLaVA-OneVision-7B (si) | 83.5 | 74.5 |
> > | LLaVA-OneVision-7B (ov) | 92.7 (+9.2) | 79.7 (+5.2) |
> >
> > *Table 2. Set-of-Mark Effectiveness Evaluation*
> >
> > | **Model** | **RefCOCOg-REC** |
> > | --- | --- |
> > | GPT4V* | 25.7 |
> > | GPT4V+SOM* | 86.4 |
> > | LLaVA-OneVision-7B (ov)** | 83.3 |
> > | LLaVA-OneVision-7B (ov) + SOM** | 90.7 (+7.4) |
> >
> > - For Video-to-Video Difference tasks, suitable benchmarks and evaluation datasets are currently limited. We will conduct and provide evaluation results when appropriate datasets become available.
> > - For visual prompting, we evaluated the model on Visual7W[1] and PointQA-LookTwice[2] to assess its understanding of complex multi-modal interactions, following the evaluation protocol of ViP-LLaVA[3]. We marked ViP-LLaVA with an asterisk (*) as it was trained on in-domain data, while our models exclude in-distribution and similar visual prompting data. Our model demonstrates strong zero-shot performance compared to ViP-LLaVA, and after training on diverse visual inputs (multi-image and video data), its performance improves further.
> > - We validated the model's robustness and generalization capabilities by comparing the original LLaVA-OneVision-7B model against a version enhanced with the SOM plugin module. While GPT4V+SOM reported performance on 100 selected images (marked with an asterisk), we evaluated on the full dataset since we couldn't access their exact test set. Nevertheless, the comparison between models with and without SOM demonstrates clear improvements, with our model showing a significant +7.4% gain in performance.
> >
> > [1] Zhu, Yuke, et al. "Visual7w: Grounded question answering in images." *Proceedings of the IEEE conference on computer vision and pattern recognition*. 2016.
> >
> > [2] Mani, Arjun, et al. "Point and ask: Incorporating pointing into visual question answering." *arXiv preprint arXiv:2011.13681* (2020).
> >
> > [3] Cai, Mu, et al. "Making large multimodal models understand arbitrary visual prompts." *arXiv e-prints* (2023): arXiv-2312.
> >
> > ---
> >
> > ### **Grammar Error in Related Works**
> >
> > **Comment:** In the second paragraph of Related Works, "For the latter, our are motivated by FLAN ..." seems to have a grammar error.
> >
> > **Response:** Thank you for catching this error. We have corrected the sentence to: *"For the latter, our design is motivated by FLAN..."*
> >
> > ---
> >
> > ### **Number of Tokens per Frame Design Choice**
> >
> > **Comment:** It seems the number of tokens per frame is 196, which is lower than the number of tokens per image in the multi-image scenarios. Are there any explanations for this design, and do experiments support the explanations?
> >
> > **Response:** The difference in token numbers arises from the design trade-offs to balance computational efficiency and information density.
> >
> > First, for video frames, the temporal dimension already introduces redundancy, and 196 tokens sufficiently capture the necessary semantic and visual details for most video tasks, if the video related Q&As are not towards the very details of the videos.
> >
> > Second, in Figure 3, we show that the necessity to balance the max tokens across image/multi-image/video tasks to better leverage model’s transferability to improve performance. As our video tasks are mostly with 32 frames, so it’s natural to use 196 tokens (4x max pool from original 729 tokens output from SO400M vision encoder).
> >
> > To validate this design choice, we conducted ablation studies comparing performance with different token counts per frame and varying frame counts.
> >
> > *Table 3. Token Configurations in Video Training and Evaluation*
> >
> > | **Model** | **Train Tokens** | **Inference Tokens** | **MVBench** | **VideoMME** | **Inf. Memory** |
> > | --- | --- | --- | --- | --- | --- |
> > | LLaVA-OneVision-7B | 196 * 32 Frames | 196 * 32 Frames | 56.7 | 58.2 | ~40GB |
> > |  | 196 * 32 Frames | 196 * 64 Frames | 58.1 | 59.5 | ~55GB |
> > |  | 196 * 32 Frames | 729 * 32 Frames | 58 | 58.6 | ~78GB, easy to OOM |
> > |  | 729 * 32 Frames | 729 * 32 Frames | 57 | 57.3 | ~78GB, easy to OOM |
> >
> > When we doubled the number of frames and total tokens, we observed performance gains (MVBench improved from 56.7 to 58.1, VideoMME from 58.2 to 59.5).
> >
> > However, increasing the per-frame tokens to 729 (by removing 2x2 pooling) did not yield the expected improvements. Even when we adjusted both training and inference to use 729 tokens, we observed decreased performance.
> >
> > These higher token configurations also substantially increased computational overhead.

---

### Decision · Action_Editor_X6qX · 2024-12-31

**Recommendation:** Accept with minor revision

**Comment:**

In this paper, the authors presented a family of large multimodal models (LMMs), termed LLaVA-OneVision, achieving strong performance across single-image, multi-image, and video tasks. A higher AnyRes strategy was proposed. Experimental results show the effectiveness of LLaVA-OV, outperforming most existing open-sourced LMMs. Transfer learning analysis was also presented.

This paper received detailed review comments from three expert reviewers. There were some concerns about the lack of details (e.g. the curated datasets, training details of the models), lack of a detailed ablation study on the key contributing components, experiments (e.g. fair comparison, small set of data,  comparison to other datasets, sensitivity, etc.), and missing discussion/comparison to related works. After the rebuttal, most of these concerns were well addressed and acknowledged by the reviewers. During the discussion phase, all reviewers agreed on the strengths of this paper, including the introduced family of LMMs, the open-source curated data, comprehensive experimental evaluation, and the strong performance. Finally, this paper received a consistent positive recommendation of 2 Leaning Accept and 1 Accept.
Considering the above, the AE is pleased to inform that this paper is accepted to TMLR, subject to the revision of including the additionally added supporting evidence (e.g. the ablation studies, experiments, discussions, etc.) during the rebuttal phase.

**Audience:**

There would be a certain amount of people in TMLR's audience be interested in knowing the findings of this paper, and actually they may benefit a group of people working in this field.

**Claims And Evidence:**

The claims made in the submission are supported by accurate, convincing and clear evidence.

---

> ### Author Response · Authors · 2025-01-29
> **Revision Update**
>
> We sincerely appreciate the valuable suggestions and feedback from the reviewers and our editor. We have carefully refined our experiments and ablation studies and updated our paper accordingly.
>
> Specifically, we have added two sub-sections in Appendix Section C to discuss “Dataset Statistics” and “Training Resource Details.” Additionally, all supplementary experiments and ablation studies have been incorporated into Section E, covering “Transferability Evaluations,” “Scaling Effects on Grids/Tokens,” and “Different Training Strategy Design.” Further insights have been included in a dedicated Discussion sub-section.

---

> > ### Comment · Action_Editor_X6qX · 2025-02-02
> > **Please use the correct TMLR format**
> >
> > Dear authors,
> >
> > Thanks for submitting your camera ready manuscript. But please use the correct TMLR format for your manuscript.
> >
> > More details please refer to the **TMLR-provided official template** (available here: https://jmlr.org/tmlr/author-guide.html) and don't alter the format. You may refer to the example papers here: https://jmlr.org/tmlr/papers/.
> >
> > Please revise the format and upload the correct version.
> >
> > Best,
> >
> > AE

---

> > > ### Comment · Action_Editor_X6qX · 2025-02-06
> > > **Please use the TMLR format and don't alter it**
> > >
> > > Please can you follow the official TMLR template and do **not** alter the format (e.g. the author list)?
> > >
> > > Please refer to example papers here: https://jmlr.org/tmlr/papers/.
> > >
> > > Also, what does the "heart" shape (next to some authors) indicate?

---

> > > > ### Author Response · Authors · 2025-02-07
> > > > **Update with correct template**
> > > >
> > > > Dear Editor,
> > > >
> > > > I sincerely apologize for the oversight regarding the author and institution details. We have now corrected them in accordance with the TMLR template. Please find the revised version attached.